# Architecture of genome-wide transcriptional regulatory network reveals dynamic functions and evolutionary trajectories in *Pseudomonas syringae*

Yue Sun[1†], Jingwei Li[1†], Jiadai Huang[1†], Shumin Li[2], Youyue Li[1], Beifang Lu[1], Xin Deng[1,3,4,5,6]*

[1]Department of Biomedical Sciences, City University of Hong Kong, Hong Kong, China; [2]The University of Hong Kong, Pokfulam, Hong Kong, China; [3]Shenzhen Research Institute, City University of Hong Kong, Shenzhen, China; [4]Tung Biomedical Sciences Center, City University of Hong Kong, Hong Kong, China; [5]Chengdu Research Institute, City University of Hong Kong, Chengdu, China; [6]Institute of Digital Medicine, City University of Hong Kong, Hong Kong, China

**\*For correspondence:**
xindeng@cityu.edu.hk

[†]These authors contributed equally to this work

**Competing interest:** The authors declare that no competing interests exist.

## eLife assessment

This work advances our understanding of transcriptional regulation of virulence and metabolic pathways in plant pathogenic bacteria. **Solid** evidence for the claims is provided by computational analysis of newly generated data on the genome-wide binding of 170 transcription factors to their target genes, together with experimental validation of the biological functions of some of these transcription factors. The findings and resources from this study will be **valuable** to researchers in the fields of systems biology, bacteriology, and plant-microbe interactions.

**Abstract** The model Gram-negative plant pathogen *Pseudomonas syringae* utilises hundreds of transcription factors (TFs) to regulate its functional processes, including virulence and metabolic pathways that control its ability to infect host plants. Although the molecular mechanisms of regulators have been studied for decades, a comprehensive understanding of genome-wide TFs in *Psph* 1448A remains limited. Here, we investigated the binding characteristics of 170 of 301 annotated TFs through chromatin immunoprecipitation sequencing (ChIP-seq). Fifty-four TFs, 62 TFs, and 147 TFs were identified in top-level, middle-level, and bottom-level, reflecting multiple higher-order network structures and direction of information flow. More than 40,000 TF pairs were classified into 13 three-node submodules which revealed the regulatory diversity of TFs in *Psph* 1448A regulatory network. We found that bottom-level TFs performed high co-associated scores to their target genes. Functional categories of TFs at three levels encompassed various regulatory pathways. Three and 25 master TFs were identified to involve in virulence and metabolic regulation, respectively. Evolutionary analysis and topological modularity network revealed functional variability and various conservation of TFs in *P. syringae* (*Psph* 1448A, *Pst* DC3000, *Pss* B728a, and *Psa* C48). Overall, our findings demonstrated a global transcriptional regulatory network of genome-wide TFs in *Psph* 1448A. This knowledge can advance the development of effective treatment and prevention strategies for related infectious diseases.

## Introduction

Transcription is a pivotal process in cellular life events. Transcription factors (TFs) play a crucial role in this process by acting as key regulators that coordinate various biological activities (*Lee and Young, 2013*; *Papavassiliou and Papavassiliou, 2016*). TFs control the recruitment of RNA polymerase by identifying and binding to the promoters of downstream genes, thereby either activating or repressing the expression of target genes (*Lambert et al., 2018*; *Wade, 2015*). Numerous studies have focused on regulatory events in eukaryotic species, such as humans (*Jolma et al., 2013*), mice (*Badis et al., 2009*), and *Saccharomyces cerevisiae* (*Zhu et al., 2009*), and prokaryotic species, such as *Escherichia coli* (*Shen-Orr et al., 2002*). However, limited comprehensive TF-binding datasets for microbial pathogens are available.

*Pseudomonas syringae*, an important Gram-negative phytopathogen and a model pathogenic bacterium, infects many plants, including economically valuable crops, resulting in substantial annual economic losses globally (*Hirano and Upper, 2000*). Upon entering host cells, *P. syringae* employs several strategies, such as changing its motility type and secreting phytotoxins, to overcome the plant's immune defences and establish colonies (*Bender et al., 1999*; *Ichinose et al., 2013*; *Taguchi and Ichinose, 2011*). *P. syringae* causes severe diseases by secreting various effector proteins through the needle-like type III secretion system (T3SS); this process is regulated by a cluster of TFs (*Cunnac et al., 2009*; *Hendrickson et al., 2000*; *Huang et al., 2022*; *Wang et al., 2018*). The alternative sigma factor RpoN activates the transcription of another alternative sigma factor, HrpL, which, in turn, binds to the pathogenicity (*hrp*) box in the promoter region of T3SS genes, regulating most of these T3SS genes (*Alfano and Collmer, 1997*; *Lan et al., 2006*; *Xiao and Hutcheson, 1994*). HrpS is one of the most important TFs that regulate numerous biological processes (*Wang et al., 2018*). Its heterodimeric complex, HrpRS, is modulated by at least 6 two-component systems (TCSs): RhpRS (*Deng et al., 2014*), CvsRS (*Fishman et al., 2018*), GacAS (*Chatterjee et al., 2003*), AauRS (*Yan et al., 2020*), CbrAB2, and EnvZ-OmpR (*Shao et al., 2021*). In particular, RhpRS serves as a master regulator of T3SS in *Psph* 1448A. RhpRS senses plant-derived signals, such as polyphenols, through the histidine kinase Pro40 within RhpS and controls the expression of T3SS genes in response to environmental stress (*Deng et al., 2014*; *Xiao et al., 2007*; *Xie et al., 2021*). Within the sensor region, the cognate response regulator RhpR undergoes modulation in its phosphorylation state by RhpS, thereby regulating a group of T3SS genes (*Deng et al., 2010*). Phosphorylated RhpR directly binds to the *hrpRS* promoter, suppressing the *hrpRS* operon and the subsequent *hrpRS-hrpL-hrp* cascade (*Deng et al., 2010*; *Deng et al., 2014*; *Deng et al., 2009*; *Shao et al., 2019*; *Xiao et al., 2007*).

Recently, through a combined analysis of RNA sequencing and chromatin immunoprecipitation sequencing (ChIP-seq), we identified seven additional TCSs (ErcS, Dcsbis, PhoBR, CzcSR, AlgB/KinB, MerS, and CopRS) that regulate the virulence of *Psph* 1448A (*Xie et al., 2022*). In addition, we developed an intricate PSTCSome (*Psph* 1448A TCS regulome) network containing numerous functional genes that respond to changing environmental conditions (*Xie et al., 2022*). Furthermore, we examined the overall crosstalk between 16 virulence-related regulators under different growth conditions, such as King's B (KB) and minimal media. By analysing differentially expressed genes and binding peaks, we constructed a *Psph* 1448A regulatory network (PSRnet), revealing the involvement of hundreds of functional genes in virulence pathways (*Shao et al., 2021*). We also elucidated the molecular mechanisms and functions of TFs binding within coding sequences (CDS) and found that CDS-binding TFs interact with cryptic promoters in coding regions, thereby regulating the expression of subgenus and antisense RNAs (*Hua et al., 2022*). We propose a luminescence reporter system designed to quantitatively measure the translational elongation rates (ERs) of T3SS-related proteins. Our findings demonstrate the key roles of transfer RNAs (tRNAs) and elongation factors in modulating translational ERs and facilitating T3SS protein synthesis (*Sun et al., 2022*).

Although many key virulence regulators in *Psph* 1448A have been studied, the global regulatory mechanism and interactions of all 301 annotated TFs across various biological processes remain unclear. To comprehensively explore the DNA-binding features and map the transcriptional regulatory network of all TFs in *Psph* 1448A, we constructed 170 TF-overexpressing strains and used ChIP-seq, a highly effective and important technology for analysing protein-DNA interactions (*Mathelier et al., 2015*). This analysis not only provided insights into the interactions between TFs and their target genes but also revealed the hierarchy (top, middle, and bottom) and co-association scores of all these TFs. We found that more than half of 270 TFs (100 TFs from HT-SELEX and 170 TFs in this

study) in downstream position tended to be regulated by top TFs and bound to the target genes with high co-associated scores. Different TF pairs were classified into 13 basic three-node submodules, including ringent loops and locked loops. In addition, we mapped the hierarchical binding network of TFs and identified three virulence-related master TFs and 23 metabolic master TFs. Furthermore, we employed ChIP-seq to determine the binding sites of five TFs in four *P. syringae* lineages (*Psph* 1448A, *Pst* DC3000, *Pss* B728a, and *Psa* C48), revealing the diversity of TF-binding events and the varying functions of TFs among different *P. syringae* strains. Topological modularity classification of the network, including TFs and target genes, revealed the diverse biological functions of TFs in *Psph* 1448A. This study provides a global and convenient platform for understanding the transcriptional regulatory characteristics and biological functions of TFs in *P. syringae*. In addition, this study provides valuable insights that can inform the development of effective therapies for not only *P. syringae* but also other associated infectious diseases.

## Results

### ChIP-seq analysis revealed the binding specificities of 170 previously uncharacterised TFs in *Psph* 1448A

Based on the current annotations available on '*Pseudomonas* Genome DB' (https://www.pseudo-monas.com/) (*Winsor et al., 2016*), we initially determined the locations of all 301 annotated TFs in the *Psph* 1448A genome (*Figure 1—figure supplement 1a*). To elucidate the binding preferences and functional characteristics of TFs of *Psph* 1448A, we performed ChIP-seq for the 170 TFs, including 3 (1.8%) predicted transcriptional regulators, 132 (77.6%) annotated transcriptional regulators, and 35 (20.6%) functional proteins with DNA-binding annotations. Based on the DNA-binding domains as annotated in the TF prediction database (*Wilson et al., 2008*), we categorised the 170 analysed TFs into 25 families (*Fan et al., 2020*). The majority of TFs belonged to the LysR, TetR, AsnC, GntR, and AraC families. Among these TFs, PSPPH4700, PSPPH3798, CysB, PSPPH1951, PSPPH4638, PSPPH3504, PSPPH3268, and Irp exhibited over 1000 binding peaks (*Figure 1—figure supplement 1b*). The enriched loci of these binding peaks indicated that these TFs displayed a significant prefer-ence for binding to promoters, directly regulating the transcription of downstream targets (*Figure 1—figure supplement 1c*). The peak loci of 10 TFs (PSPPH0286, PSPPH0411, PSPPH0711, PSPPH1734, PSPPH2357, PSPPH2407, PSPPH2862, PSPPH3155, PSPPH3431, PSPPH3468, PSPPH4127, PSPPH4622, and PSPPH4768) were completely enriched in the promoter region, with the majority of them belonging to the LysR family. Taken together, the 170 tested TFs in *Psph* 1448A had over 26,000 DNA-binding peaks distributed across different regions of target genes, suggesting their direct regu-latory functions.

### Hierarchical TFs reflected multiple higher-order network structures

Transcriptional changes in bacteria are often manipulated by a complex network of TFs. However, bacterial TFs usually have been studied individually or in small clusters with related functions. To comprehensively investigate the associations of all TFs in *Psph* 1448A at a system level, we constructed a hierarchical network of 270 analysed TFs (100 TFs from HT-SELEX and 170 TFs in this study). The findings revealed 1757 TF interactions among these 26,000 binding events (*Supplementary file 1a*). Subsequently, we computed information flow parameters for each TF (*Gerstein et al., 2012*). In brief, we defined out-degree (O) and in-degree (I) as the number of interactions of a TF in the hierarchical network, representing the regulation of other factors by this TF and the regulation of this TF by other factors, respectively. The difference between O and I indicated the direction of information flow in the network. Hierarchy height (H) was defined as the normalised metric of information circulation, calcu-lated as H=(O−I)/(O+I). When H was close to 1 (H ≈ 1), it indicated that these TFs tended to regulate other factors and occupy upstream positions in the network. Conversely, when H was close to −1 (H ≈ −1), it indicated that these TFs were more likely to be regulated than to regulate other TFs, occupying downstream positions in the network. Based on these criteria, we categorised the 270 analysed TFs (100 TFs from HT-SELEX and 170 TFs in this study) into three levels: 54 (20%) executive TFs (such as AlgQ, LexA2, and PSPPH0222) at the top level, 62 (23%) communicat ive TFs (such as MexT, PsrA, and PSPPH1100) at the middle level, and 147 (54%) foreman TFs (such as PobR, DksA2, and PSPPH0755) at the bottom level (*Figure 1a*, *Supplementary file 1b*). The presence of a larger number of TFs (147)

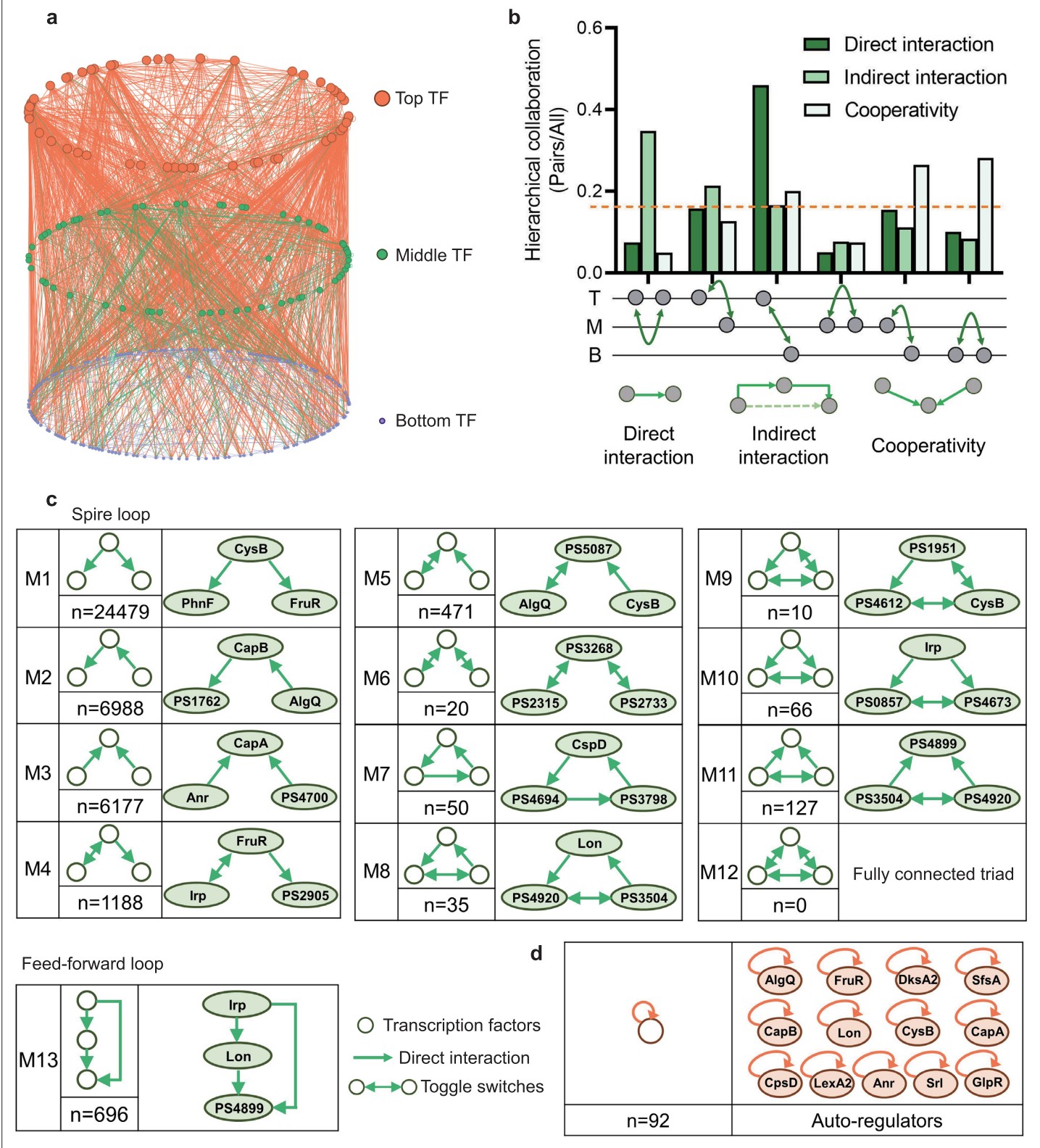

**Figure 1.** Hierarchical height and collaboration of transcription factors (TFs) reveal the multiple regulatory patterns in *Psph* 1448A. (**a**) Close-up representation of 262 TFs hierarchy in *Psph* 1448A (8 TFs showed no hierarchical characteristic). Nodes depict TFs. Colours of edges represent source-bases. (**b**) Enrichment of different collaborating (direct interaction, indirect interaction, and cooperativity) TF pairs at top (T), middle (M), and bottom (B) levels. We defined indirect interaction if two TFs co-associated with one target DNA. Cooperative TF pair was defined if their common target is from

*Figure 1 continued on next page*

*Figure 1 continued*

a TF. Grey nodes below the graph represent TFs. The dashed orange line indicates the averaged level of collaboration. (**c**) Thirteen 3-node submodules with the number of occurrences and an example. Spire loop is the most enriched submodule. Edges represent the regulatory direction. (**d**) Auto-regulations are accompanied by the number of occurrences and 13 auto-regulators as examples.

The online version of this article includes the following figure supplement(s) for figure 1:

**Figure supplement 1.** Summary of chromatin immunoprecipitation sequencing (ChIP-seq) results in *Psph* 1448A.

**Figure supplement 2.** Graph diagram of feedforward loop in *Psph* 1448A.

at the bottom level indicated a high degree of information flow, suggesting the maximisation of the number of downward-pointing edges in the network.

Different numbers of TFs in three levels in the hierarchical network revealed that more than half of TFs (bottom-levels TFs) tend to be regulated by other TFs and then directly bound to target genes. This tendency indicated a downward information flow of transcription regulation of *P. syringae*. Therefore, we defined the direct binding between two TFs as a direct interaction and investigated collaborations within and between hierarchy levels, specifically intra-level (top to top 'TT', middle to middle 'MM', and bottom to bottom 'BB') and inter-level (top to middle 'TM', top to bottom 'TB', and middle to bottom 'MB') interactions, indirect interaction if two TFs co-associated with one target DNA and cooperative TF pair was defined if their common target was from a TF (*Figure 1b*, *Supplementary file 1a*). In terms of the top-level TFs, direct interactions became more enriched as the hierarchy level of their collaborators decreased. Direct interactions between TB pairs constituted the most substantial portion, accounting for nearly half of all interactions. A similar pattern was observed among the bottom-level TFs, where interactions diminished as the hierarchy level of their collaborators decreased. Compared with interactions among the top- and bottom-level TFs, middle-level TFs, serving as information transmission centres, exhibited lower levels of intra-level collaborations. In summary, transcriptional regulation within the TF hierarchy was predominantly manipulated by top-level TFs, which directed the flow of information to downstream TFs.

## Multiple three-node submodules revealed the regulatory diversity of TFs in the *Psph* 1448A regulatory network

Natural networks, including transcriptional regulation networks, usually show complex characteristics (*Newman, 2001*; *Strogatz, 2001*). Among complex networks, some small-scale networks demonstrate numerous connections between individual information nodes and information clusters (*Amaral et al., 2000*; *Jeong et al., 2000*). To investigate the basic structural features of our transcription network, we defined directed edges as direct interactions between two TF nodes and identified global submodules comprising different TF nodes. In this study, we specifically focused on three-node modules, which were considered as 'network motifs' (*Milo et al., 2002*). Using algorithms designed to detect recurring modules (*Shen-Orr et al., 2002*), we scanned our hierarchy network and identified 40,307 different pairs across 13 basic three-node submodules (*Figure 1c*).

In the first six submodules, we observed that two TF nodes established a relationship only through another node. We denoted these submodules (M1–M6) as 'ringent loops'. These seemingly simpler regulation modules appeared more in the *Psph* 1448A transcriptional regulatory network, especially the first module (M1, n=24,479), indicating that *Psph* 1448A favours the use of simple but efficient modes for modulating transcriptional regulation. For example, PhnF and FruR were directly regulated by CysB (M1), and CapA was coregulated by Anr and PSPPH4700 simultaneously (M3, n=6177). In addition, M6 (n=20) contained pairs of mutually regulating TFs (toggle switches), such as PSPPH3268, PSPPH2315, and PSPPH2733.

The remaining seven submodules, denoted as 'locked loops' (M7–M13), comprised subordinate three-node regulatory modules within our network. Notably, no instances of a 'fully connected triad' (M12) were observed in our network (n=0). We found 50 'self-loop' submodules (M7) in our network, including CspD, PSPPH4694, and PSPPH3798, which engaged in mutual regulatory interactions. Among these submodules, M9 (n=10) was the least common and contained six pairs of toggle switches involving mutually regulating TFs (such as PSPPH1951, PSPPH4612, and CysB), which were similar to those found in the human transcriptional regulatory network (*Gerstein et al., 2012*). Notably, the most enriched locked loop in our network was M13 (n=696), denoted as a 'feedforward loop', which

has been extensively studied in other species such as humans and *E. coli*. In this submodule, upstream TFs regulated targeted TFs either by binding directly or manipulating other TFs (*Figure 1—figure supplement 2*). For example, TF Irp directly controlled PSPPH4899 and also indirectly regulated it by binding to Lon.

Taken together, the simplest and most effective submodule M1 and the coregulatory submodule M13 played crucial roles in the transcriptional regulation of TFs in *Psph* 1448A. In addition, we found 92 auto-regulators in our hierarchy network. These auto-regulators are important and always act as repressors to maintain the cellular stability under different environmental stimuli (*Alon, 2007*; *Becskei and Serrano, 2000*; *Lee et al., 2002*). For example, LexA and CysB as negative auto-regulators were indicated to reduce cell-to-cell fluctuations in the steady-state level of the TF (*Becskei and Serrano, 2000*; *Rosenfeld et al., 2002*). These regulators are regarded as bistable switches that further influence the expression of downstream genes (*Burda et al., 2011*). For example, our previous study demonstrated that Lon is a dual-function regulator involved in the regulation of virulence and metabolism in *Psph* 1448A (*Hua et al., 2020*). Lon was identified as an auto-regulator in this study. Furthermore, DksA2, which is widely regarded as a protective protein against oxidative stress (*Fortuna et al., 2022*), was identified as a new bistable switch in this study. In summary, the classification of the TF hierarchy and the identification of enriched network modules not only offer functional predictions for transcriptional regulators but also provide insights into the communication network that governs TF regulation in *Psph* 1448A.

## High co-association pairs occurred more in bottom-level TFs in *Psph* 1448A

In addition to direct interactions between TFs, we found a notable preference for co-binding peaks among different TF pairs. Briefly, we counted overlapping regions within the binding peaks of all TF pairs. The ratio of intersection regions to the union set of all peaks between two TFs was identified as the genome-wide co-association of specific TFs. We first analysed co-association scores between TF pairs and grouped the scores into three TF pairs (TT, MM, and BB). The majority of TFs tended to cooperate with other TFs and co-bind to specific genomic regions (*Figure 2—figure supplement 1a*, *Figure 2—figure supplement 2a* and *Figure 2—figure supplement 3a*). The dark distribution in BB pair indicated that high co-association scores preferred to occur in bottom-level TFs. To identify the potential functions of TFs in each level, we used the target genes to perform functional annotations using hypergeometric tests (BH-adjusted $p < 0.05$) based on gene sets derived from the Gene Ontology (GO) and Kyoto Encyclopaedia of Genes and Genomes (KEGG) databases (*Figure 2—figure supplement 1b*, *Figure 2—figure supplement 2b*, *Figure 2—figure supplement 3b*, *Supplementary file 2a, c, and d*). The functional categories of TFs at these three levels encompassed various regulatory pathways. For example, the top-level TF PSPPH4700 was involved in siderophore transportation and phosphorelay signal transmembrane transportation. The middle-level TF Lon regulated GTP binding and ribosome transcription. The bottom-level TF PSPPH3486 was involved in amino acid transportation, PSPPH0101 in ATP binding, and PSPPH1049 in catalytic activity. Notably, we found that TFs at the top level, without cooperating TFs, exhibited a large number of binding peaks (*Figure 2—figure supplement 1a*). Previous study suggested that TFs preferred to regulate target genes by recruiting to specific sites of other TFs, facilitating the direct binding between other TFs and their specific targets, a phenomenon defined as tethered binding (*Wang et al., 2012*). Therefore, top TFs in our study more likely regulated target genes by 'direct binding' than 'tethered binding'. For example, the top-level TF PSPPH4700 yielded over 1700 peaks but cooperated with only 24 top-level TFs with low co-association scores of about 0.05 (*Supplementary file 2b*).

To further elucidate this pattern, we examined the 125 TFs that were analysed through ChIP-seq and exhibited high co-association scores. We determined the co-association patterns among these regulators (*Figure 2a*) and classified them into four clusters, denoted as C1–C4. Notably, C1 and C4 contained higher proportions of bottom-level TFs. Based on the analysis in *Figure 1b*, we found that the proportions of bottom-level TF interaction in all the TF pair interactions and direct interaction were 43% and 49%, respectively. These results indicated that the bottom-level TFs tended to regulate downstream genes through cooperating with other level TFs. When comparing the co-association scores of TF pairs, we observed a stronger tendency towards cooperation among lower-level TFs, especially bottom-level TFs. In particular, 35 bottom-level TFs in C4 (83%) exhibited coregulation

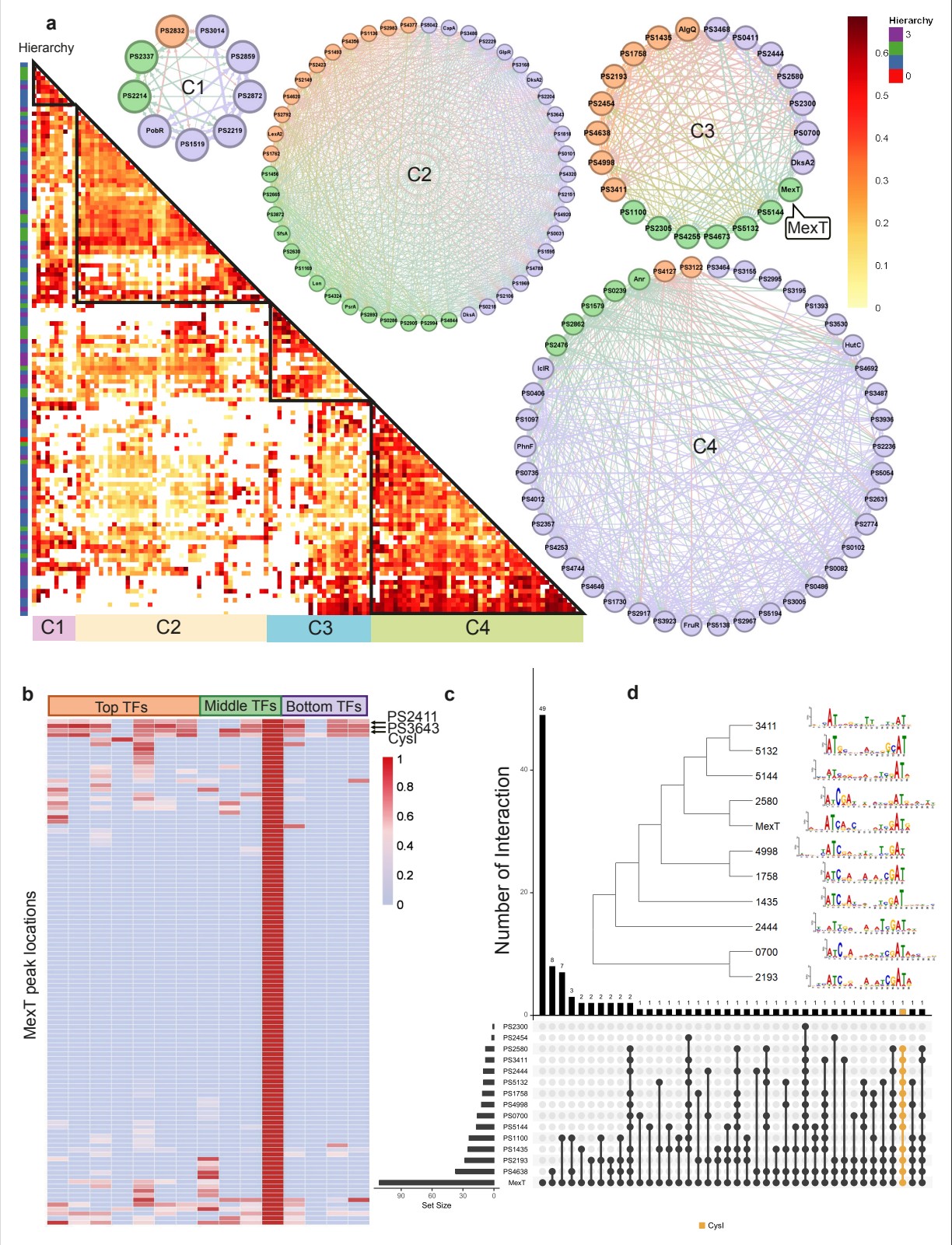

**Figure 2.** Bottom transcription factors (TFs) share the same binding sequences to coregulate in *Psph* 1448A. (**a**) The co-association map for 170 TFs in *Psph* 1448A shows the co-associated scores of binding peaks of these TFs (rows) that overlap each TF peak (columns). The three-coloured rectangles represent three different TF levels. C1–C4 represent four clusters of TFs according to the co-associated scores. The TFs in corresponding cluster are shown in the circle diagram. Orange nodes represent top TFs. Green nodes represent middle TFs. Purple nodes represent bottom TFs. The colours

*Figure 2 continued on next page*

*Figure 2 continued*

of edges are the mixture of two-source TF colours. (**b**) The heat-map of MexT indicates the associated scores of binding peaks of TFs in C3 (columns) that overlap the binding peaks (rows) of MexT. PSPPH2411, PSPPH3643, and CysI are the top 3 TFs with high associated scores. (**c**) UpSet plot shows the number of genes uniquely targeted TFs or co-targeted by multiple TFs in C3. The vertical black lines represent shared TF-binding sites. The y axis represents the number of overlapped binding sites across the linked TFs. The x axis represents the number of binding sites for each TF. Orange line represents the most enriched gene *cysI* that is co-targeted by 12 TFs in C3. (**d**) Motifs of MexT and other 10 TFs in C3 which show similar binding sequences.

The online version of this article includes the following figure supplement(s) for figure 2:

**Figure supplement 1.** Co-association and virulence-related functional category of transcription factors (TFs) at top level.

**Figure supplement 2.** Co-association and virulence-related functional category of transcription factors (TFs) at middle level.

**Figure supplement 3.** Co-association and virulence-related functional category of transcription factors (TFs) at bottom level.

at the same peak locations with high co-association scores. For instance, PhnF and PSPPH4692 co-bound to three target genes (PSPPH4117, PSPPH4216, and PSPPHB0021), with co-association scores as high as 0.92. This co-binding relationship between all TF pairs was defined as an indirect interaction. In contrast to the trend observed for direct interactions and cooperativity, we observed stronger correlations between top-level TFs and TFs situated at higher levels (top and middle levels; *Figure 1b*). For example, the top-level TFs PSPPH3798 and CysB co-bound to the promoter region of *flhB*, which encodes a flagellar biosynthesis protein. We found a similar co-association pattern among bottom-level TFs. However, middle-level TFs showed the weakest internal correlations, suggesting that they tend to collaborate with TFs from other levels rather than with other middle-level TFs. For instance, we observed that the middle-level TF PSPPH1100 cooperated with top-level TF PSPPH3798 and bottom-level TF PSPPH4920, binding to the promoter region of *hopG1*, which encodes a type III effector.

We found that the correlation between bottom-level TFs was weaker in C3 than in the other clusters. To further explore the DNA-binding characteristics of all TFs within the same cluster, we investigated the peak features of the LysR family TF MexT in C3 as an example. The peak locations in target genes of seven top-level TFs (PSPPH1435, PSPPH1758, PSPPH2193, PSPPH2454, PSPPH4638, PSPPH4998, and PSPPH3411), three middle-level TFs (PSPPH1100, PSPPH5132, and PSPPH5144), and four bottom-level TFs (PSPPH0700, PSPPH2300, PSPPH2444, and PSPPH2580) were compared with MexT. MexT showed higher co-association scores (more than 60 scores) with more top-level TFs. The analysis of the peak locations of MexT demonstrated that MexT showed closer co-association relationships with top-level TFs (*Figure 2b*). Within the binding peaks of MexT, the target genes PSPPH3643 (LysR family TF), PSPPH2618 (sulphite reductase (NADPH) haemoprotein, CysI) and PSPPH2411 (hypothetical protein) displayed high co-association scores with other TFs. UpSet plots demonstrated that 12 TFs in C3 bound to a highly overlapping region of CysI (*Figure 2c*). These results suggest that these genes were prioritised for coregulation by MexT. To further explore the binding features in C3, we determined the motifs of 11 TFs based on their binding sequences obtained via ChIP-seq using MEME (*Bailey et al., 2009*), even though the binding motif of MexT was previously investigated (*Tian et al., 2009*). We identified a 15 bp consensus motif (ATN11AT) throughout the 11 analysed TFs (*Figure 2d*), demonstrating a high degree of consensus in co-association patterns among TFs in *Psph* 1448A.

## Virulence-associated pathways were primarily regulated by top-level TFs in *Psph* 1448A

Because the pathogenicity of *P. syringae* mainly depends on T3SS and other virulence-associated pathways (*He, 1998*), we particularly focused on TFs that bind to numerous virulence-related genes. Seven pathways manipulate the virulence of *Psph* 1448A (*Huang et al., 2022*; *Shao et al., 2019*), namely T3SS, biofilm production, motility, nucleotide-based secondary messenger function, quorum sensing (QS), phytotoxin production, and siderophore production. To comprehensively investigate the transcription regulatory mechanism underlying the virulence of *Psph* 1448A, we calculated the hierarchical heights of TFs involved in virulence regulation and the virulence genes modulated by them. This analysis provided insights into the organisation of the virulence regulatory network in *Psph* 1448A, where virulence-involved TFs were categorised into three tiers (*Figure 3a*).

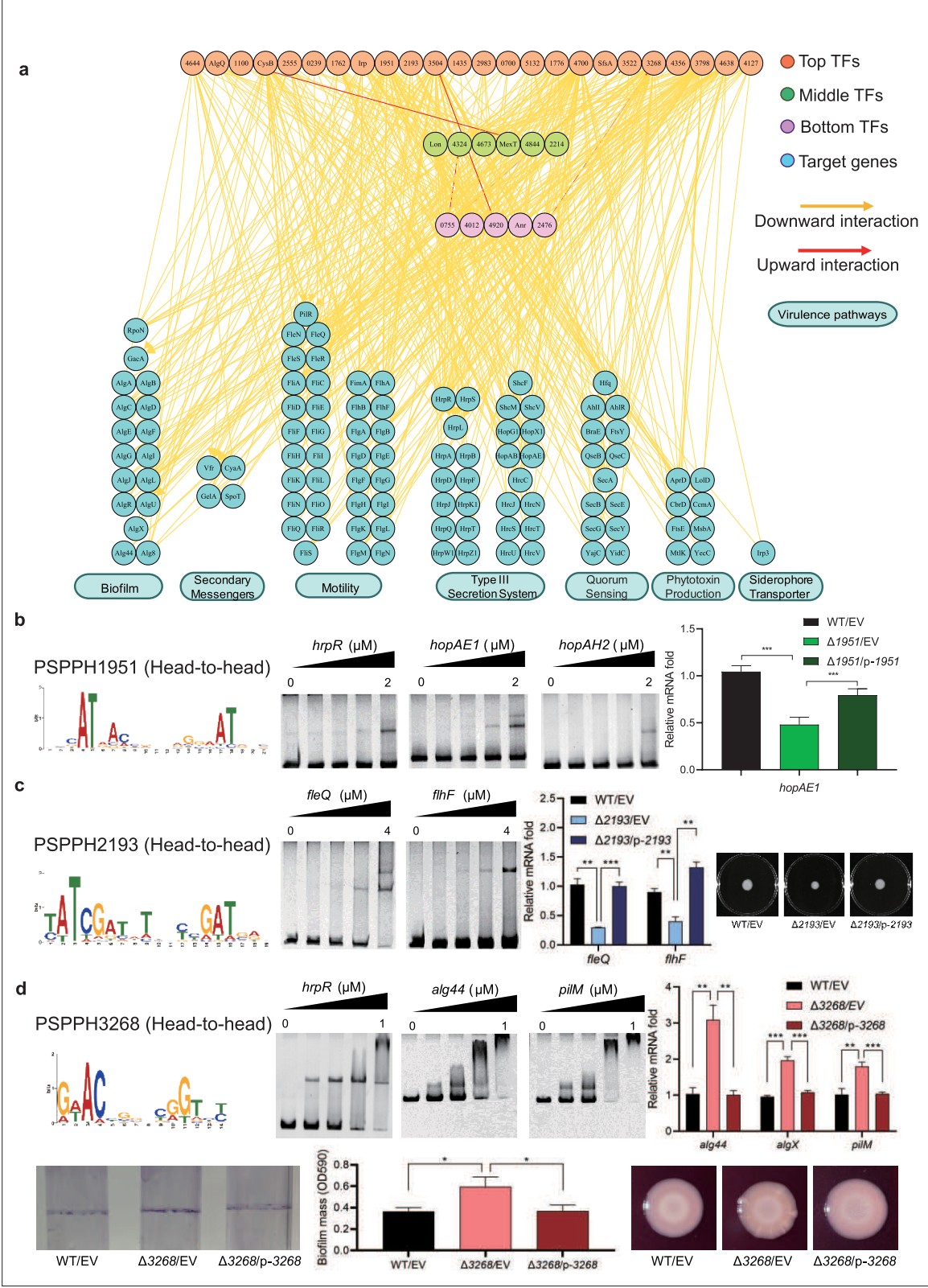

**Figure 3.** Virulence hierarchical regulatory network reveals 35 transcription factors (TFs) involved in virulence. (**a**) Virulence hierarchical regulatory network shows the TF hierarchy and the large pool of target genes of multi-TF. Target genes are related with seven key virulence pathways, including biofilm formation, secondary messengers, motility, T3SS, quorum sensing (QS), phytotoxin production, and siderophore transporter. Orange nodes represent top TFs. Green nodes represent middle TFs. Purple nodes represent bottom TFs. Blue nodes represent target genes. Yellow edges represent

*Figure 3 continued*

downward point. Red edges represent upward point. (**b**) The head-to-head binding motif of PSPPH1951, the validation of the binding sites of PSPPH1951 by electrophoretic mobility shift assay (EMSA), and the detection of expression of target gene *hopAE1* in wild-type (WT), ΔPSPPH1951, and complementary strain by real-time quantitative PCR (RT-qPCR). The validated binding sites are from the promoters of the *hrpR*, *hopAE1*, and *hopAH2*. (**c**) The binding motif of PSPPH2193 is head-to-head. EMSA confirms the direct binding of PSPPH2193 to the promoters of *fleQ* and *flhF*. RT-qPCR confirms the positive regulation of PSPPH2193 on the expression of *fleQ* and *flhF*. Motility assay validates the weaker motility of ΔPSPPH2193 than WT and complementary strain. (**d**) The binding motif of PSPPH3268 is head-to-head. EMSA confirms the direct binding of PSPPH3268 to the promoters of *hrpR*, *alg44*, and *pilM*. RT-qPCR confirms the negative regulation of PSPPH3268 on the expression of *alg44*, *algX*, and *pilM*. Crystal violate staining assay and the quantification of biofilm formation validate the negative regulation of PSPPH3268 on the biofilm formation. Congo red assay confirms the negative regulation of PSPPH3268 on colony morphologies and extracellular polysaccharide (EPS) production. Student's t-test. n.s., not significant, *p≤0.05, **p≤0.01, and ***p≤0.001.

The online version of this article includes the following source data and figure supplement(s) for figure 3:

**Source data 1.** PDF file containing original electrophoretic mobility shift assay (EMSA), motility, biofilm, and extracellular polysaccharides (EPS) detection for *Figure 3b–d*, indicating the relevant binding and phenotypes.

**Source data 2.** Original electrophoretic mobility shift assay (EMSA) gels and photos for EMSA, motility, biofilm, and extracellular polysaccharides (EPS) detection displayed in *Figure 3b–d*.

**Figure supplement 1.** The validation of the binding sites of virulence-related transcription factors (TFs) in *Psph* 1448A.

**Figure supplement 1—source data 1.** PDF file containing original electrophoretic mobility shift assay (EMSA) for *Figure 3-figure supplement 1a–c, e, and g*.

**Figure supplement 1—source data 2.** Original electrophoretic mobility shift assay (EMSA) gels corresponding to *Figure 3-figure supplement 1a–c, e, and g*.

We found three transcriptional regulatory channels governing virulence regulation in *Psph* 1448A. The first channel was the direct trigger, which has been extensively studied in previous studies and is referred to as the 'one-step trigger' here (**Fan et al., 2020**). These TFs are recognised as master regulators that directly respond to biological events without additional intermediaries (**Chan and Kyba, 2013**). In our previous study, we identified TrpI, RhpR, GacA, and PSPPH3618 as master regulators in T3SS and 16 master regulators in other virulence pathways (**Fan et al., 2020**). In line with this definition, we recognised 35 TFs (PSPPH4644, AlgQ, PSPPH1100, CysB, PSPPH2555, PSPPH0239, PSPPH1762, Irp, PSPPH1951, PSPPH2193, PSPH3504, PSPPH1435, PSPPH2983, PSPPH0700, PSPPH5132, PSPPH1776, PSPPH4700, SfsA, PSPPH3522, PSPPH3268, PSPPH4356, PSPPH3798, PSPPH4638, PSPPH4127, Lon, PSPPH4324, PSPPH4673, MexT, PSPPH4844, PSPPH2214, PSPPH0755, PSPPH4012, PSPPH4920, Anr, and PSPPH2476) that participate in various virulence pathways. More than 68% of these TFs (24 of 35 TFs) were at the top levels, indicating that the virulence of *Psph* 1448A is primarily regulated by top-level TFs. Among these TFs in the network, the top-level TFs PSPPH1951, PSPPH2193, and PSPPH3268 were found to have abundant virulence-associated target genes based on ChIP-seq results. The *de novo* motif analysis of their peak sequences revealed the presence of three head-to-head motifs: a 17 bp motif (AT-N13-AT) for PSPPH1951, a 15 bp motif (ATC-N9-GAT) for PSPPH2193, and a 10 bp motif (AC-N6-GT) for PSPPH3268 (*Figure 3b–d*).

## Three master TFs were identified to participate in virulence

To further verify the biological functions of these three uncharacterised TFs, we first purified the TF proteins and performed an electrophoretic mobility shift assay (EMSA) to confirm their direct interactions with key virulence genes *in vitro*. Next, we generated TF deletion strains to detect the transcription levels of target genes. We first detected the effect of the mutants on bacterial growth. The growth curve assay showed that the deletion of TF genes had no significant influence on the *Psph* growth (*Figure 3—figure supplement 1-d*). We found that PSPPH1951 directly regulated multiple T3SS genes, including *hrpR*, *hopAE1* and *hopAH2* (*Figure 3b*). Among them, the expression of *hopAE1* in ΔPSPPH1951 was significantly decreased by twofold compared with the WT, suggesting that PSPPH1951 acts as an activator of *Psph* 1448A T3SS (*Figure 3b*). In addition, PSPPH1951 was found to bind to the promoters of type IV pili genes (*pilG*, *pilF*, and *pilZ*; *Figure 3—figure supplement 1*).

*Pst* DC3000 enhances its ability to infect the host by increasing bacterial motility (**Buell et al., 2003**). ChIP-seq data (*in vivo*) and EMSA (*in vitro*) results showed that PSPPH2193 interacted with the promoters of motility-related genes, such as *fleQ* (encoding a flagellar regulator) and *flhF* (encoding

a flagellar biosynthesis regulator; *Figure 3c*). Real-time quantitative PCR (RT-qPCR) results indicated that *fleQ* and *flhF* were downregulated fourfold in ΔPSPPH2193 compared with the WT. As expected, ΔPSPPH2193 exhibited weaker motility than the WT and complementary strain in KB medium (*Figure 3c*), indicating that PSPPH2193 serves as an activator of motility in *Psph* 1448A. To further investigate the influence of PSPPH2193 on the pathogenicity of *P. syringae*, we performed the plant infection assay. As expected, we found that the bacterial numbers of ΔPSPPH2193 strain were significantly reduced compared with WT strain (*Figure 3—figure supplement 1f*). In summary, PSPPH2193 regulates the pathogenicity of *Psph* 1448A through motility pathway by directly binding to target genes.

PSPPH3268 was found to influence the pathogenicity of *Psph* 1448A by regulating multiple virulence-related pathways. During the initial stage of *P. syringae* infection, the bacteria produce biofilm components, including extracellular polysaccharides (EPSs), type IV pili, and other highly viscous compounds. These components help the bacteria to establish colonies, providing protection against the host's immune defences and antimicrobial agents (*Whitchurch et al., 2002*). We found that PSPPH3268 strongly interacted with the promoters of key genes involved in biofilm production such as *hrpR*, the alginate biosynthesis gene *alg44*, and the type IV pilus assembly gene *pilM* (*Figure 3d*). Furthermore, the transcription levels of *alg44*, *algX* (encoding the alginate biosynthesis protein), and *pilM* were markedly enhanced in ΔPSPPH3268. This resulted in enhanced biofilm formation and EPS production when the PSPPH3268 gene was deleted and then restored when PSPPH3268 was expressed (*Figure 3d*). These results demonstrated that PSPPH3268 acts as a master regulator in various virulence-related pathways. In addition, we identified that the TF PSPPH3798 binds to the promoters of flagellar-related genes (*fliK*, *fliE*, *fliD*, and *fleQ*; *Figure 3—figure supplement 1g*).

In addition to the 'one-step trigger' mechanism, we found that TFs also regulate downstream genes through one or two other TFs at different levels, which were regarded as 'one jump-point trigger' and 'two jump-point trigger'. For example, PSPPH2555 indirectly influenced biofilm formation (*algD*), motility (*fleQ*), T3SS (*hrpR*), QS (*ahlR* and *secE*), and phytotoxin production (*aprD*) by directly regulating the bottom-level TF PSPPH4920 (*Supplementary file 3*). PSPPH3504 was found to be involved in a PSPPH3504-Lon/PSPPH4324/PSPPH4844-PSPPH0755/PSPPH4920/PSPPH4012-target gene pathway (*Figure 3—figure supplement 1h*, *Supplementary file 3*, '/' represents sibling nodes and '–' represents downward regulation). Among these TFs, PSPPH0755, PSPPH4920, and PSPPH4012 are considered key performer TFs because they mediate most of the transcription regulatory signals from multiple TFs. We also found reverse regulatory pathways in our network. For example, the middle-level TF PSPPH4673 was found to directly regulate the top-level TF PSPPH4700 and then indirectly control the transcription of many virulence-related genes through regulating the bottom-level TFs PSPPH0755, PSPPH4920, Anr, and PSPPH2476 (*Supplementary file 3*).

## Systematic mapping of TF targets revealed key metabolic regulators in *Psph* 1448A

In addition to enhancing pathogenicity and resisting host defences, *P. syringae* adjusts its metabolic activities to survive in unpredictable environments (*Rico et al., 2011*). To comprehensively understand metabolic regulation in *Psph* 1448A, we constructed a hierarchical network that includes key regulators and the genes they trigger, similar to the virulence hierarchical network. We focused on eight metabolic pathways, namely amino acid biosynthesis, DNA replication, ATP-binding cassette (ABC) transportation, oxidative phosphorylation, tricarboxylic acid (TCA) cycle, RNA polymerase, phosphonate metabolism, and methyl-accepting chemotaxis (*Figure 4a*). Compared with the virulence network shown in *Figure 3A*, we identified more TFs involved in metabolic regulation, many of which exhibited numerous interactions with genes related to oxidative phosphorylation (178 binding peaks) and the TCA cycle (154 binding peaks; *Figure 4—figure supplement 1*). In a previous study, 12 master regulators were reported to control various metabolic pathways, including LexA1, PSPPH3004, and PSPPH1960, involved in reactive oxygen species resistance (*Fan et al., 2020*). RhpR participates in several metabolic pathways, such as ABC transporters and oxidoreductase activity (*Shao et al., 2019*). Lon is involved in glucokinase and oxidoreductase activity (*Hua et al., 2020*). MgrA, GacA, PilR, PsrA, RpoN, CvsR, OmpR, and CbrB2 participate in oxidation resistance, amino acid transportation, and other metabolic pathways (*Shao et al., 2021*). Here, we identified 111 TFs regulating these eight metabolic pathways. Similar to the aforementioned virulence network, three transcriptional regulatory

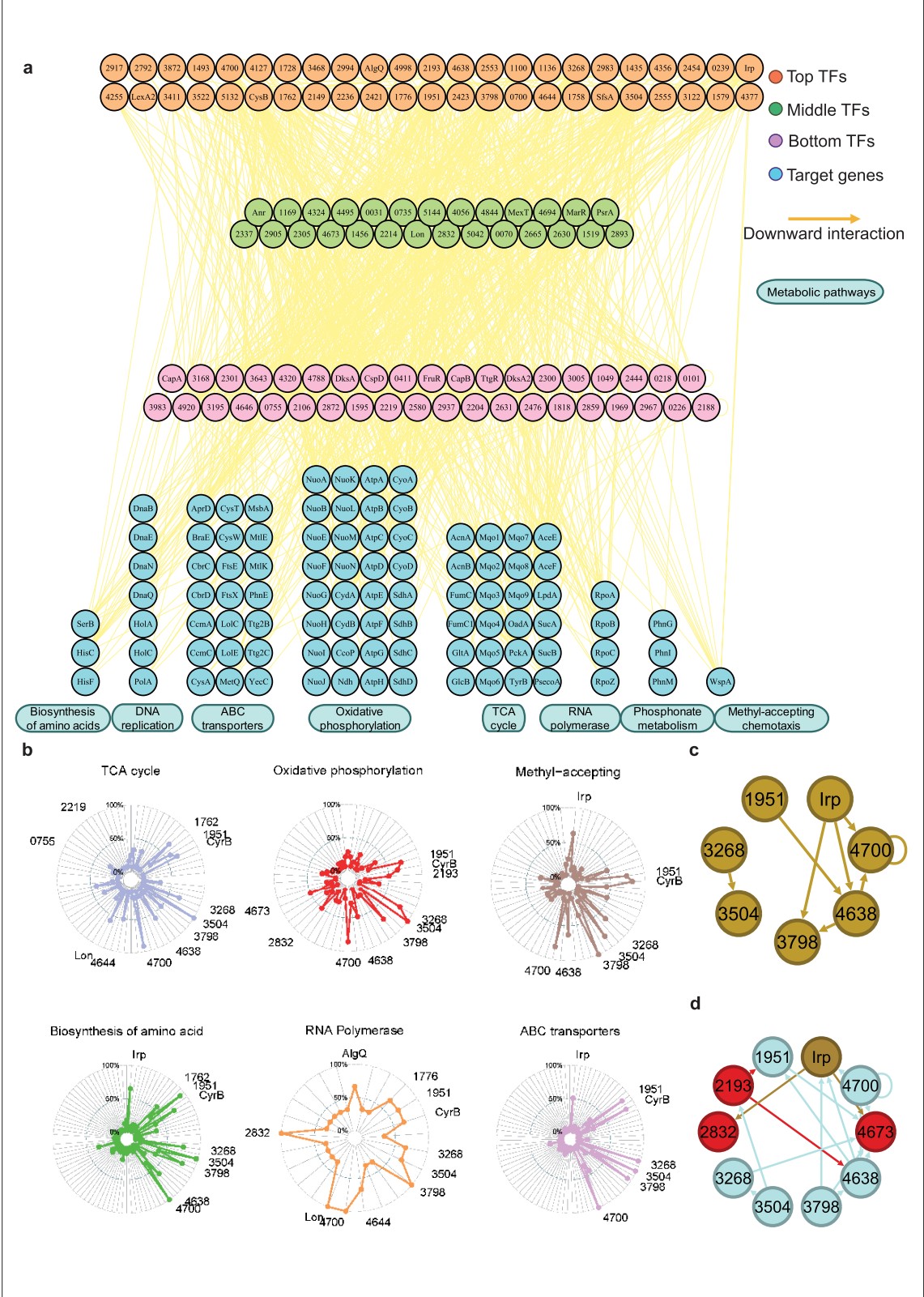

**Figure 4.** Hundreds of transcription factors (TFs) are identified to participate in metabolic pathways. (**a**) Metabolic hierarchical regulatory network shows the TF hierarchy and the large pool of target genes of multi-TF. Target genes are related with eight key metabolic pathways, including biosynthesis of amino acids, DNA replication, ATP-binding cassette (ABC) transporter, oxidative phosphorylation, tricarboxylic acid (TCA) cycle, RNA polymerase, phosphonate metabolism, and methyl-accepting chemotaxis. Orange nodes represent top TFs. Green nodes represent middle TFs. Purple nodes

*Figure 4 continued on next page*

*Figure 4 continued*

represent bottom TFs. Blue nodes represent target genes. Yellow edges represent the direct interaction. (**b**) Radar plots show the putative key regulators identified in six different metabolic pathways, including TCA cycle, oxidative phosphorylation, methyl-accepting, biosynthesis of amino acids, RNA polymerase, and ABC transporter. Each radiation line represents a key regulator, and the radial length of the thick coloured line is the rate of target genes to the associated genes, representing the significance of the enrichment of the TF-target genes within each pathway. (**c**) TFs involved in the methyl-accepting pathway bound to the promoters of TFs in the same category. Brown nodes represent the TFs that are responsible for methyl-accepting pathway. The brown arrows point to the targeted TFs. (**d**) TFs involved in the oxidative phosphorylation pathway bind to the promoters of TFs in the methyl-accepting pathway. Red nodes represent the TFs that are responsible for oxidative phosphorylation pathway. Brown nodes represent the TFs that are responsible for methyl-accepting pathway. Blue nodes represent the TFs involved in these two pathways. The arrows point to the targeted TFs and the arrow colours are source-based.

The online version of this article includes the following source data and figure supplement(s) for figure 4:

**Figure supplement 1.** Metabolic functional category of transcription factors (TFs) at three different levels.

**Figure supplement 2.** Key transcription factors (TFs) in different metabolic pathways.

**Figure supplement 2—source data 1.** PDF file containing original electrophoretic mobility shift assay (EMSA) for *Figure 4-figure supplement 2b–d*.

**Figure supplement 2—source data 2.** Original electrophoretic mobility shift assay (EMSA) gels corresponding to *Figure 4-figure supplement 2b–d*.

channels were observed for the metabolic pathways. To provide a detailed view of metabolic regulation in *Psph* 1448A, we counted the number of functionally annotated genes related to each pathway and calculated the proportion of targets for each TF, highlighting key regulators in these eight metabolic pathways using radar plots (*Figure 4b* and *Figure 4—figure supplement 2a*).

We found that the TFs CysB and PSPPH3268 regulate all eight metabolic pathways, whereas the TFs PSPPH1951, PSPPH3798, PSPPH3504, and PSPPH4700 were predicted to regulate seven metabolic pathways. Notably, the TF PSPPH0755 was found to bind to the promoters of PSPPH5210 (encoding ATP synthase F0F1 subunit delta) and PSPPH3109 (encoding the NADH dehydrogenase subunit A NuoA). The monomer motif (CTGAA) of PSPPH0755 was identified through MEME analysis. The interactions in these two metabolic pathways were confirmed through EMSA (*Figure 4—figure supplement 2b*). The TF PSPPH3798 was predicted to bind to the promoters of genes in two pathways, including PSPPH3881 (encoding the methyl-accepting chemotaxis protein WspA) and PSPPH5119 (encoding the phosphate transport system regulatory protein PhoU). The 15 bp binding motif of PSPPH3798 was determined to have a head-to-head orientation (ATCG-N7-CGAT). EMSA results confirmed these interactions (*Figure 4—figure supplement 2c*). In addition to the TF PSPPH3798, the TF PSPPH4638 had a binding site in the promoter region of the PSPPH3881 gene. PSPPH4638 was also predicted to interact with PSPPH0550 that encodes phosphoserine phosphatase SerB. The 8 bp monomer motif (ATTTTCAA) of PSPPH4638 was identified, and the binding interactions were confirmed through EMSA (*Figure 4—figure supplement 2d*).

In yeast, TFs in a functional category appear to bind to genes in the same category (*Simon et al., 2001*), and we observed a similar pattern in *Psph* 1448A. For example, TF Irp, a key regulator in the methyl-accepting pathway, bound to the promoters of PSPPH3798, PSPPH4638, and PSPPH4700, which were also identified as key regulators in the same pathway (*Figure 4c*). In addition, we found that TFs from different categories often bound to the promoters of TFs responsible for other cellular processes. For example, key regulators controlling oxidative phosphorylation (highlighted in red and blue; PSPPH1951, PSPPH2193, PSPPH2832, PSPPH3268, PSPPH3504, PSPPH3798, PSPPH4638, PSPPH4673, and PSPPH4700) bound to TFs playing key roles in the methyl-accepting pathway (highlighted in brown and blue; PSPPH1951, PSPPH3268, PSPPH3504, PSPPH3798, PSPPH4638, PSPPH4700, and Irp; *Figure 4d*). These results demonstrate that many regulatory processes are often achieved through coregulation by a series of multifunctional TFs throughout the network, enabling *Psph* 1448A to coordinate transcriptional regulation processes across multiple cellular processes.

## TFs indicated large functional variability across different pathovars in *P. syringae*

Although TF functions exhibit both inter- and intra-species variability, most previous studies on TFs have focused on the molecular mechanism of a single strain (*Galardini et al., 2015*). To investigate the regulatory mechanism of TFs across different strains of *P. syringae*, we selected four model strains: *P. syringae* pv. *syringae* 1448A, *P. syringae* pv. *tomato* DC3000, *P. syringae* pv. *syringae* B728a, and *P.*

*syringae* pv. *actinidiae* C48. We used the genome of 1448A as a reference and conducted a homology analysis of 1448A protein sequences with those of the other three strains (***Supplementary file 4***). We determined a high proportion of homologous proteins in the three strains (4983 in *Pst* DC3000, 4982 in *Pss* B728a, and 4984 in *Psa* C48; ***Supplementary file 4***). Across the four strains, all 301 annotated TFs were present (***Supplementary file 5***). We selected five TFs (Irp, PSPPH2193, PSPPH3122, PSPPH4127, and OmpR) to construct TF-overexpressing strains in *Pst* DC3000, *Pss* B728a, and *Psa* C48 before performing ChIP-seq analysis. We identified the binding sites of all these TFs in the four strains and found divergent binding preferences for the same TFs in different strains. Most target genes of each TF in one or two strains were unique. In particular, Irp bound to 19 target genes that were conserved in all four tested strains, including *purB* (encoding adenylosuccinate lyase), *cceA2* (encoding the chemotaxis sensor histidine kinase), and *gidA* (encoding the tRNA uridine 5-carboxymethylaminomethyl modification protein). Four highly conserved target genes were also found to directly interact with PSPPH4127. Evolutionary analysis of the binding peaks of TFs suggested high binding specificity and varying levels of conservation of these TFs in the tested strains (***Figure 5a***).

In addition to the intersection between *Psph* 1448A and *Psa* C48, we observed differences between the target genes of all five TFs (***Figure 5b***) and the peak locations (TF-target interactions; ***Figure 5c***) in these four pathovars. The inconsistency between the number of targets and peaks suggested that some target genes were regulated by at least one different TF in these four strains, which is similar to regulation in *Pseudomonas aeruginosa* (***Trouillon et al., 2021***). To confirm the presence of target genes regulated by the same TF or different TFs in various strains, we compared the peak locations of *gidA* and *rpoD* (RNA polymerase sigma factor) as two examples. TF Irp was found to bind to the promoter of *gidA* in four strains (***Figure 5d***) and had 19 conserved target genes in these strains (***Supplementary file 4***). In addition, PSPPH4127 had four conserved target genes (*rpoD*, PSPPH1001, PSPPH1998, and PSPPH_5016) in all four strains (***Figure 5e***). These results showed that Irp and PSPPH4127 exhibited higher functional conservation than the other three TFs. We also found that PSPPH2193 in 1448A and Irp in *Pst* DC3000 bound to the promoter of *rpoD* (***Figure 5e***). Differences in the regulation of the same targets by different TFs were also observed in more than 1500 target genes, suggesting the potential diversity of the transcriptional regulation of TFs in our network.

Furthermore, we performed motif and GO analysis to investigate the functional characteristics of TFs (PSPPH3122 and PSPPH4127) in *P. syringae* strains (***Figure 5—figure supplements 1–2***). The similar motifs but in different target genes were found in various TFs in four *P. syringae* strains. For PSPPH3122, motif ($AGACN_4GATCAA$) and motif ($CGGACGN_3GATCA$) were found in *Psph* 1448A and *Pst* DC3000 strains, respectively, which showed similar binding motifs of $GGACGN_3GATCA$ (***Figure 5—figure supplement 1a and b***). However, the similar motif was distributed in different target genes, such as PSPPH3091 (encoding recombination factor protein RarA) in *Psph* 1448A and PSPTO1535 (encoding translation elongation factor Ts) in *Pst* DC3000, respectively. Functional enrichment analysis of target genes of PSPPH3122 indicated its relationship with recombinase activity and DNA recombination in 1448A strain, while relating with other functions in other strains, such as structural constituent of ribosome in *Pss* B728a and *Pst* DC3000 strains (***Figure 5—figure supplement 1c***). We also found four motifs of PSPPH4127 in four *P. syringae* strains, which showed a core binding sequence of GCCA (***Figure 5—figure supplement 2a–d***). Functional enrichment analysis showed its relationship with recombinase activity in *Psph* 1448A strain, and RNA binding, structural constituent of ribosome, translation, and ribosome in *Pss* B728a strain (***Figure 5—figure supplement 2e***). These results showed a highly functional variability of TFs in *P. syringae*. In this point, we suggested that these TFs possessed similar binding specificities in different *P. syringae* strains, which located in various regions of target genes, resulting in diversification of transcriptional regulation.

## Topological modularity of the transcription regulatory network exhibited various functions in biological processes in *Psph* 1448A

Complex networks in nature often exhibit topological and/or functional modularity (***Dittrich et al., 2008***; ***Olesen et al., 2007***). To explore the modularity in the TF-binding network, we used a partitioning algorithm (with a resolution of 0.9) to classify network elements into different subsets using Gephi. TFs were the primary nodes, and each edge represented the regulatory flow of TFs to target genes, which were small nodes. Our analysed TFs and their target genes were divided into 16 modules, each represented by a different colour (***Figure 6a***). Multiple application modulatory can

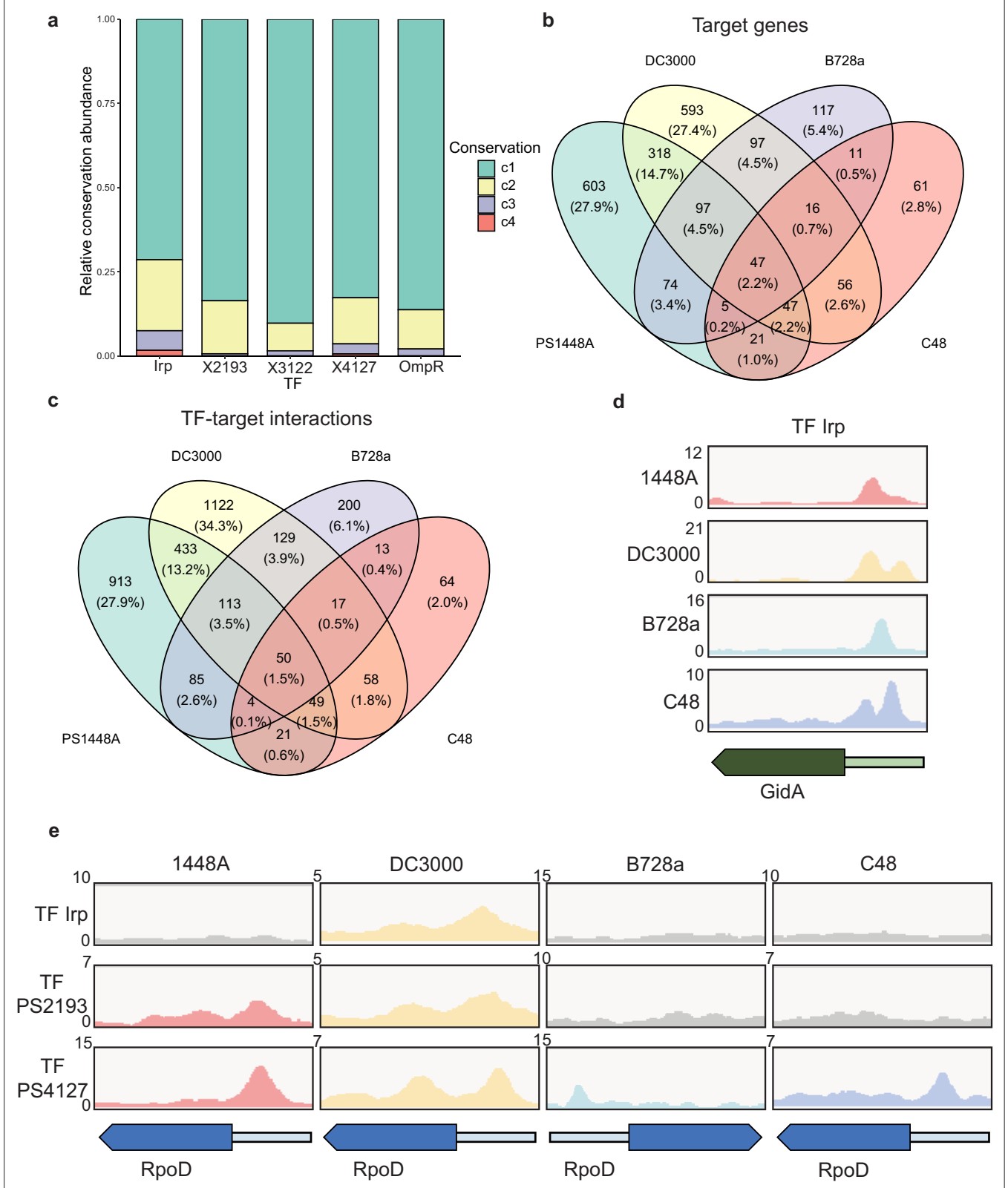

**Figure 5.** Various conservations are observed in transcription factors (TFs) between different *P. syringae* pathovars. (**a**) Proportion of the TF-target genes detected in one, two, three, or four tested genomes. C1–C4 represent the conservation of targets in one, two, three, and four strains. (**b–c**) Repartition of the total pool of target genes (**b**) or TF-target interactions (**c**) in four tested strains. (**d**) Enrichment coverage tracks of chromatin immunoprecipitation sequencing (ChIP-seq) against negative controls for TF Irp with binding sites on the promoter of *gidA* in all four genomes. (**e**) Enrichment coverage

*Figure 5 continued on next page*

*Figure 5 continued*

tracks of ChIP-seq against negative controls (grey tracks) for the three TFs (Irp, PSPPH2193, and PSPPH4127) with binding sites on the promoter of *rpoD* in four tested strains.

The online version of this article includes the following figure supplement(s) for figure 5:

**Figure supplement 1.** Motif and functional enrichment analysis of PSPPH3122 in *Psph* 1448A, *Pss* B728a, and *Pst* DC3000 strains.

**Figure supplement 2.** Motif and functional enrichment analysis of PSPPH4127 in *Psph* 1448A, *Pss* B728a, *Pst* DC3000, and *Psa* C48 strains.

be achieved depending on different clustering resolution. In our case, a resolution of 0.9 provided moderate modularity and ensured that each module contained 2.7–12.1% of the total elements and exhibited correlations with each other. We found that almost all nodes in the network had connections both within and between modules, indicating that the 16 modules were not isolated and contributed to extensive information flow throughout the network to regulate transcription in *Psph* 1448A (**Figure 6b**). Module 12 appeared to play a central role in facilitating large information flows with other modules. Module 15 also exhibited transcriptional information exchanges both between modules and within the same module.

Among the 16 modules, Module 2 was involved in most nodes (443 elements, including 41 TFs and 402 target genes). Module 3 contained the least TFs (181 elements, including 2 TFs and 179 target genes). To investigate the potential correlation between topological modularity and biological functions, we performed GO term and KEGG pathway enrichment analysis for each module by hypergeometric test (BH-adjusted p<0.05). As expected, 15 modules were enriched in specific biological functions (**Figure 6c and d**). In some modules, hundreds of elements were assessed. For example, genes in Module 2 (443 nodes), Module 12 (327 nodes, including 22 TFs and 318 target genes), and Module 14 (432 nodes, including 3 TFs and 324 target genes) were mainly enriched in the regulation of transcription and DNA binding (**Figure 6c**). DksA, a TF in Module 2, played a key role in regulating transcription-coupled DNA repair (**Meddows et al., 2005**) and also participated in oxidative phosphorylation, amino sugar and nucleotide sugar metabolism and RNA degradation in *E. coli*. Twenty TFs (such as CapA, CapB, CysB, FruR, and MarR) classified in Module 12 were identified to be involved in transcriptional regulation. The TF PSPPH3798 located in Module 14 was observed to be involved in flagellar assembly, and these interactions were confirmed by EMSA (**Figure 3—figure supplement 1g**). The genes in Module 4 were enriched in oxidoreductase activity, and our analysis revealed that the TF PSPPH0755 played a role in regulating the oxidative phosphorylation pathway (**Figure 6d**). MexT in Module 5, which was previously associated with motility in *P. syringae* pv. *tabaci* 6605 (**Kawakita et al., 2012**), was found to participate in biofilm formation and QS pathways in our study. In addition, we not only identified the crucial roles of the TFs PSPPH1951 (Module 6), PSPPH2193 (Module 11), and PSPPH3268 (Module 15) in T3SS pathways, bacterial motility, and biofilm formation, respectively (**Figure 3b–d**), but also reported their potential biological functions in aminoacyl-tRNA biosynthesis (PSPPH1951), RNA degradation (PSPPH2193), and catalytic activity (PSPPH3268) (**Figure 6d**). Our results allowed us to identify the potential regulators of specific pathways and perform functional predictions for hypothetical proteins in *Psph* 1448A. For instance, PSPPH1503 in Module 15, which encodes a hypothetical protein, was possibly correlated with glycerophospholipid metabolism.

## Discussion

Most microbial studies on genome-wide transcriptional regulatory network focus on *S. cerevisiae* and *E. coli*, which reveal the principles of architecture and interactions of their regulatory networks. The analysis of the transcription regulatory associations in *S. cerevisiae* mainly relies on the databases such as YEASTRACT (YEAst Search for Transcriptional Regulators And Consensus Tracking) (**Teixeira et al., 2018**). In *E. coli*, relative complete transcriptional regulatory network has been generated through integrating three different data sources (RegulonDB, Ecocyc, and TRN-SO) (**Ma et al., 2004**). However, few study has yet comprehensively evaluated TFs in other prokaryotic species throughout a genome (**Ishihama et al., 2016**). In this study, we successfully generated the most complete transcriptional regulatory network and data source, which profiled the transcriptional regulatory features of both the aforementioned 100 TFs (**Fan et al., 2020**) and an additional 170 TFs in *Psph* 1448A through ChIP-seq. By mapping the TF-target hierarchical regulatory networks, we identified several

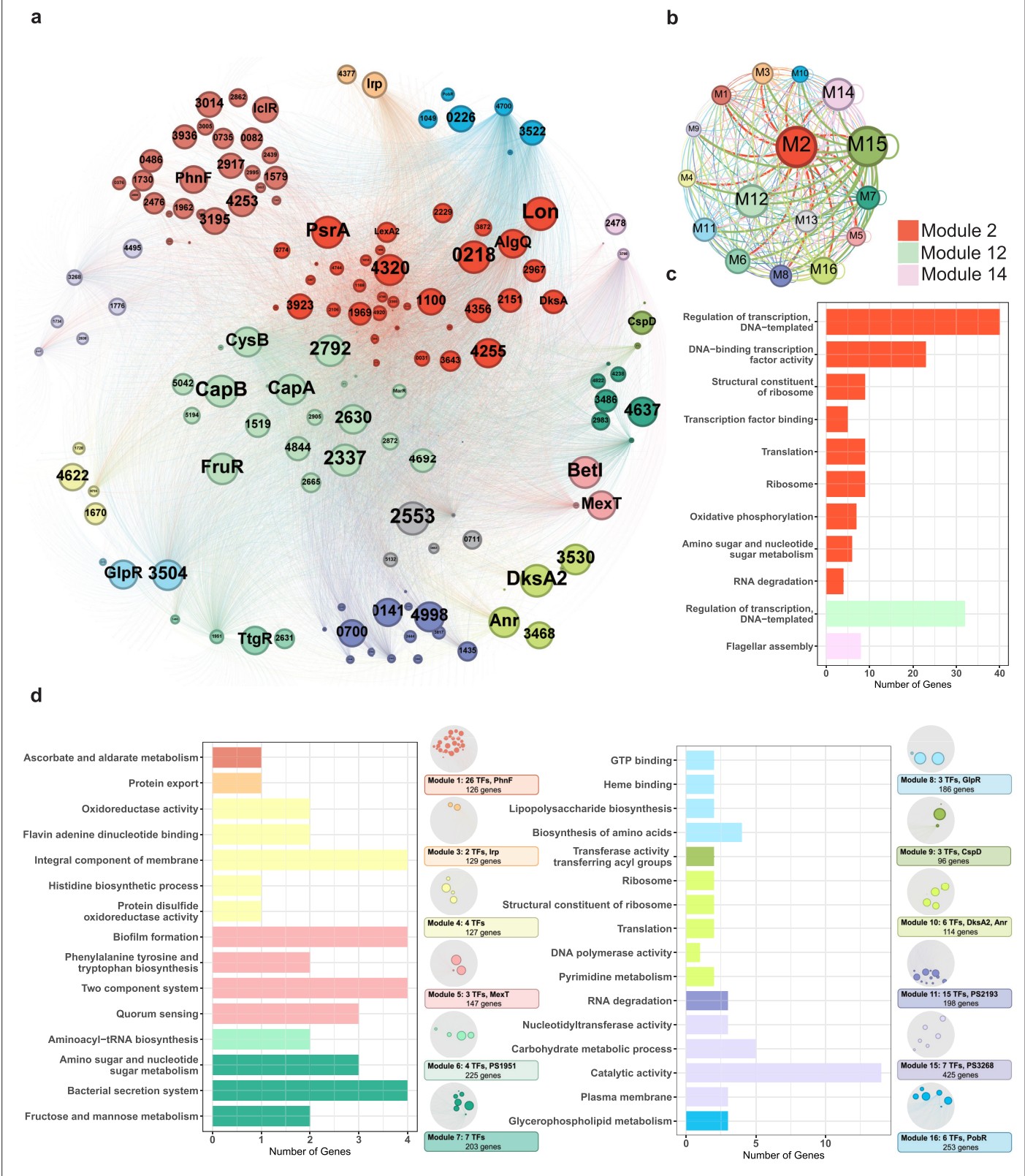

**Figure 6.** Functional modularised regulatory network in *P. syringae* exhibits the specific functions of both transcription factors (TFs) and their target genes. (**a**) The functional categorical regulation network in *Psph* 1448A analysed by Gephi (resolution 0.9). The 16 modules (both TFs and their target genes) are labelled in different colours. TF nodes are shown as corresponding sized circles representing their expression level. Their target genes are shown as corresponding-coloured dots in the background. TF-target edges are shown as corresponding-coloured lines between nodes. (**b**) Graph

*Figure 6 continued on next page*

*Figure 6 continued*

diagram indicates the connections between TF and their targets in modules. Module nodes are shown as corresponding-coloured circles with size proportional to the number of nodes within. Edge colours are source-based, and edge thicknesses represent the connected quantity between modules. (**c–d**) Functional category enrichment analysis of genes in each module, p<0.05.

novel master regulators involved in significant biological processes. Furthermore, our evolutionary analysis and assessment of the topological functional modularity of TFs and their respective targets revealed the evolutional conservation and functional diversity of TFs in *P. syringae*.

Although transcriptional regulatory networks are considered conserved (*Perez and Groisman, 2009*), many studies reveal highly functional variability of TFs in inter- and intra-species (*Galardini et al., 2015*). These observed diversities between different strains of the same species mainly result from the expression levels of TFs, contents of target genes, and differences of binding sequences (*Trouillon et al., 2021*). In our study, we observed large differences of DNA-binding characteristic of TF Irp between the C48 strain and the *Pst* DC3000 strain. The functional diversity of TFs may arise from the large difference in the contents of target genes and TFs, which are regarded as the main determinant of transcriptional regulatory (*van Duin et al., 2023*), although Irp display high homology in these pathovars.

Collaborations between TFs at higher levels (top and middle) were enriched, a pattern similar to the tendencies observed in human TFs (*Gerstein et al., 2012*). In particular, TT TF pairs exhibited a greater degree of cooperative gene regulation, whereas TB TF pairs accounted for nearly half of direct interactions within all communications. Furthermore, we observed that both direct physical regulation and cooperative interactions were the least common among MM TF pairs. By contrast, in humans, direct regulation tends to occur between TT or TM TF pairs. Furthermore, interactions between TFs, in any form, within human and yeast transcriptional regulatory networks are more likely to appear between middle TFs, which act as information transfer centres (*Bhardwaj et al., 2010*). This may be attributed to the higher abundance of bottom-level TFs than higher-level TFs observed in prokaryotic microorganisms, a pattern also found in *E. coli* (*Bhardwaj et al., 2010*). This finding indicates that the bottom-level TFs that are more likely to be regulated are evolutionarily preferred in multicellular eukaryotic organisms. However, they found that the deficient bottom-level TFs are more commonly co-associated with other same-level TFs in *E. coli*. When comparing co-associated TF pairs and the cooperativity of TFs, we observed a distinct and inverse relationship in *Psph* 1448A compared with yeast or *E. coli* (*Figure 1B*). The enriched cooperativity of bottom-level TFs with high co-associated scores indicated that these bottom-level TFs preferred to coregulate target genes by binding to the same peak locations. Notably, seven TFs without correlations with other TFs appeared to independently participate in biological processes. These findings not only shed light on the inherent properties of direct regulation and co-association across various species but also indicate the unique characteristics of *Psph* 1448A in response to dynamic environmental variations.

The fundamental units of a transcriptional regulatory network are positive and negative loops. For a more comprehensive description, these regulatory units can be classified into six submodules, namely auto-regulation, multicomponent loops, feedforward loops, single-input, multi-input, and regulator chain (*Lee et al., 2002*). In yeast, only 10 TFs were found to auto-regulate themselves, whereas the majority of regulatory units among 116 TFs in *E. coli* exhibit auto-regulation (*Thieffry et al., 1998*). Similarly, we identified 92 auto-regulators in our transcriptional regulatory network in *P. syringae*. Feedforward loops are highly prevalent in eukaryotic transcriptional regulatory networks, such as human and yeast (*Gerstein et al., 2012*; *Lee et al., 2002*). The 696 feedforward loops (M13) identified in our study also highlighted this cooperative regulation of TFs in response to small signals in prokaryotic species, such as *P. syringae*. This enriched submodule is regarded as a temporal switch that provides constant feedback to respond rapidly to signal impacts (*Shen-Orr et al., 2002*). Multi-step regulation assists master regulators in enhancing the initial information flow (*Goldbeter and Koshland, 1984*). Unlike M3, the most abundant module (868) in the human TF regulatory network, we observed that M1 was the most prevalent module (24,479) in the *Psph* 1448A transcriptional regulatory network. This finding indicates that *Psph* 1448A prefers achieving transcriptional regulation through few TFs to ensure rapid transmission and response to environmental signals.

Despite more than 20 years of research, our understanding of the global regulatory network in the plant pathogen *P. syringae* remains inadequate. Our previous studies reported 16 key virulence-related

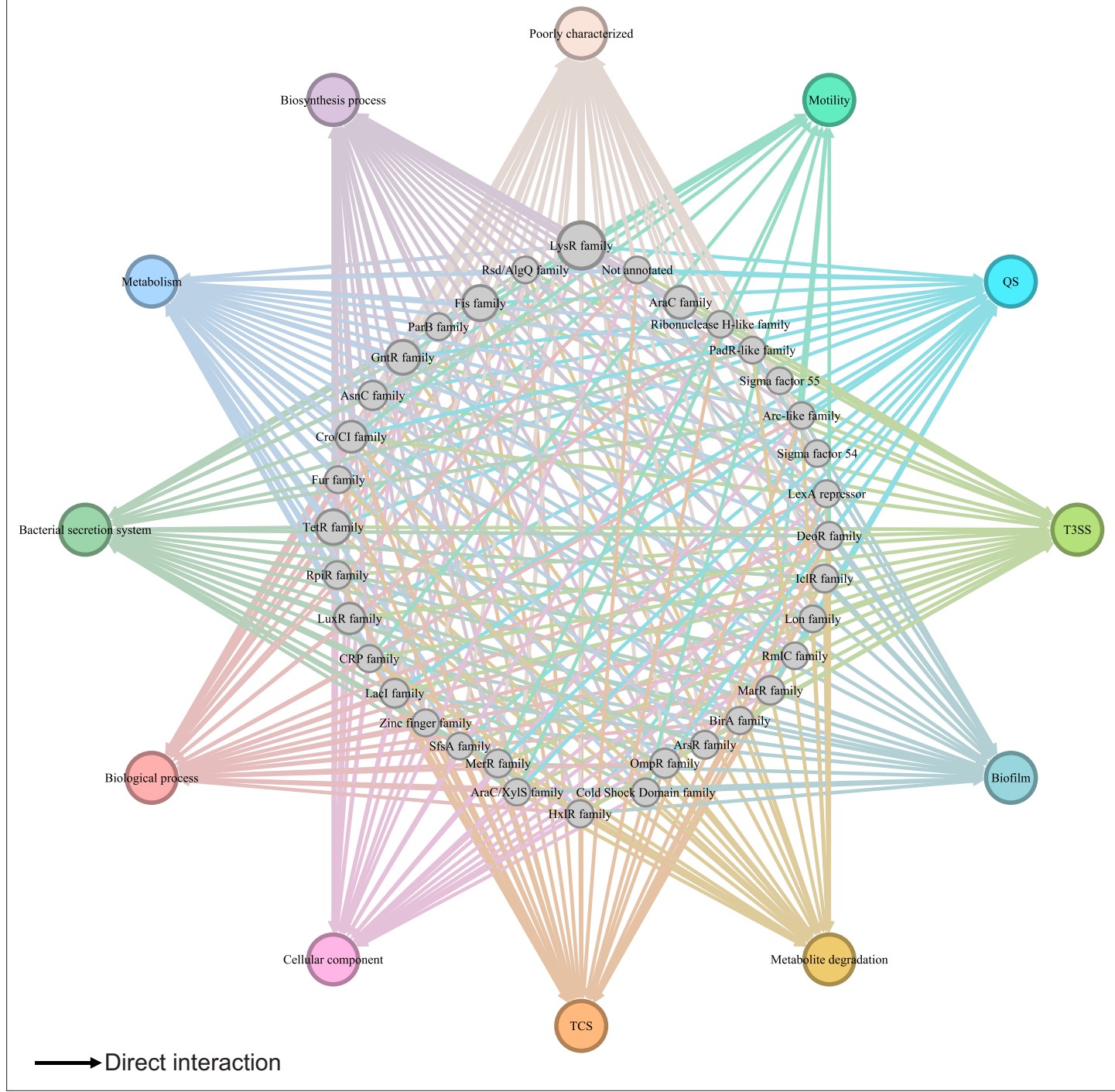

**Figure 7.** Global transcriptional regulatory network in *Psph* 1448A. The integrated transcriptional regulatory network reflects the interactions between all transcription factors (TFs) classified into 39 families from different DNA-binding domain types and target genes annotated from pathway annotations. The targets are shown in 12 pathways with various colours. TF nodes are grey as corresponding sizes representing TF number in the family. Edge colours are target-based.

regulators (*Shao et al., 2021*), 25 master virulence-related regulators (*Fan et al., 2020*), and 7 global regulators acting as TCSs (*Xie et al., 2022*). However, the regulatory relationships and functional crosstalk among all TFs in *P. syringae* remain unknown. In this study, we integrated all available inter-action information concerning almost all TFs in *Psph* 1448A and mapped the first comprehensive transcriptional regulatory network in this plant pathogen (*Figure 7*). This network offers a global

view of the multiple functions of TFs in *Psph* 1448A. We also identified 35 vital TFs that participate in virulence pathways and 111 key TFs involved in metabolic pathways across the global transcriptional regulatory network in *Psph* 1448A. This analysis uncovered new functions of previously characterised TFs, such as MexT. In addition to the previously reported *mexEF-oprN* operon (*Sawada et al., 2018*), this study identified *fleQ* (a flagellar regulator) and *shcF* (a type III chaperone protein) as the targets of MexT. In addition, we investigated the functional evolution and potential intra-species variability of TFs in *P. syringae,* demonstrating the functional diversity of TFs among *P. syringae* species during their long course of evolution. Based on the aforementioned results, we established the *Psph* 1448A transcriptional regulatory network (PSTRnet) database, which contains detailed binding peak information for all TFs in *Psph* 1448A (https://jiadhuang0417.shinyapps.io/PSTF_NET/). This database serves as a valuable platform for presenting, searching, and downloading regulatory information of transcription in *Psph* 1448A.

We observed crosstalk between not only TFs but also various pathways. For example, the TF PSPPH4700 directly regulated *fleQ* and *hrpR* and indirectly regulated *fleQ* and *hrpR* through the PSPPH4700/PSPPH4324/PSPPH0755 cascade. Feedforward loop modules are usually coherent, meaning that the direct effect of downstream TFs has the same regulatory direction (positive or negative) as the indirect effect of upstream TFs (*Shen-Orr et al., 2002*). This observation enhanced our understanding of the influence of TFs in the network on their target genes based on the identified effects of other TFs. Furthermore, the TF PSPPH4700 was identified to bind to the promoters of genes involved in seven metabolic pathways, namely the TCA cycle, oxidative phosphorylation, methyl-accepting, amino acid biosynthesis, RNA polymerase, ABC transportation, and DNA replication. Our results proved the possibility of crosstalk between different pathways in *Psph* 1448A.

Taken together, our study provides comprehensive insights into the DNA-binding characteristics and potential regulatory pathways of almost all annotated TFs in *Psph* 1448A. The global transcriptional regulatory network can not only contribute to the development of novel drugs to combat *P. syringae* infections but also advance research on the molecular mechanisms of TFs in other pathogens.

## Materials and methods
### Bacterial strains, culture media, plasmids, and primers
The bacterial strains, plasmids, and primers used in this study are listed in **Supplementary file 6**. *P. syringae Psph* 1448A, *Pst* DC3000, *Pss* B728a, and *Psa* C48 strains were grown at 28°C in KB medium shaking at 220 rpm or KB agar plates (*King et al., 1954*). *E. coli* BL21(DE3) or DH5α strains were grown at 37°C in Luria-Bertani (LB) medium shaking at 220 rpm or on LB agar plates. Antibiotics used for *P. syringae Psph* 1448A, *Pst* DC3000, and *Pss* B728a wild-type (WT) strains and mutants were rifampicin at 50 μg/ml; *P. syringae Psph* 1448A, *Pst* DC3000, and *Pss* B728a overexpression strains were rifampicin at 50 μg/ml and spectinomycin at 50 μg/ml, C48 overexpression strains were spectinomycin at 50 μg/ml; *P. syringae* 1448A strains with pK18mobsacB plasmids for mutant construction were rifampicin at 50 μg/ml and kanamycin at 50 μg/ml. Antibiotics used for *E. coli* with pET28a plasmids for protein purification, and pK18*mobsacB* plasmids were kanamycin at 50 μg/ml; *E. coli* with pHM1 plasmids for overexpression strain construction were spectinomycin at 50 μg/ml.

### Construction of overexpression strains and mutants
Overexpression strains and mutants of *P. syringae* were constructed as previously described (*Kvitko and Collmer, 2011*; *Shao et al., 2021*). In brief, for overexpression strain, the open reading frame *Caillet et al., 2019* of each TF-coding gene was amplified by PCR from *P. syringae* genome and cloned into pHM1 plasmid. The ligated fragments were inserted into the digested pHM1 plasmids (*Hind*III) using ClonExpress MultiS One Step Cloning Kit (Vazyme). The recombinant plasmids were transformed into the *P. syringae* WT strain. The single colonies were confirmed by western blot. Over-expressed TF strains may cause artificial DNA-binding, and also contribute to exploring further bindings of TFs which may not be found at physiological levels. To avoid the artificial binding of TFs, we used EMSA and RT-qPCR to further verify the analysed results, which ensured TFs at physiological levels. For mutants, the upstream (~1500 bp) and downstream (~1000 bp) fragments of TF ORF were amplified by PCR from the *Psph* 1448A genome and digested with *Xba*I respectively, and then ligated by T4 DNA ligase (*Jumper et al., 2021*). The ligated fragments were inserted into the digested

pK18*mobsacB* plasmids (*Xba*I) using ClonExpress MultiS One Step Cloning Kit (Vazyme). The recombinant plasmids were transformed into the *Psph* 1448A WT strain. The single colonies were selected from the sucrose plates and further screened for two kinds of KB plates (with kanamycin/rifampin and with rifampin) concurrently. The single colonies losing kanamycin resistance were further verified by RT-qPCR to detect the mRNA level of corresponding TF genes.

## ChIP-seq analyses

*Psph* 1448A was used for our main ChIP-seq. As per previous description (*Blasco et al., 2012*), the overexpression strains of corresponding TFs and *P. syringae* WT with empty pHM1 plasmid was cultured in 30 ml KB medium to $OD_{600}$=0.6. Bacterial cultures were cross-linked with 1% formaldehyde for 10 min at 28°C and then the reaction was stopped by the addition of 125 mM glycine for 5 min. The centrifugated bacteria were washed twice with Tris Buffer (20 mM Tris-HCl [pH 7.5] and 150 mM NaCl) and washed again with IP Buffer (50 mM HEPES-KOH [pH 7.5], 150 mM NaCl, 1 mM EDTA, 1% Triton X-100, 0.1% sodium deoxycholate, 0.1% SDS, and mini-protease inhibitor cocktail). The centrifugated bacteria were preserved at –80°C or continued for the next experiments. The bacteria were subjected with IP Buffer and then sonicated to pull down the DNA fragments (150–300 bp). The supernatant was the DNA-TF-HA tag complex and used as IP samples. IP experiments and control sample were incubated with agarose-conjugated anti-HA antibodies (Sigma) in IP Buffer. The complex of DNA-TF-anti-HA agarose was applied to washing, crosslink reversal, and purification (*Blasco et al., 2012*). The 150–250 bp DNA fragments were selected for library construction. The libraries were sequenced using the HiSeq 2000 system (Illumina). Two biological replications have been performed for all ChIP-seq experiments. ChIP-seq reads were mapped to the *P. syringae Psph* 1448A genome (NC_005773.3), *Pst* DC3000 (NC_004578.1), *Pss* B728a (NC_007005.1), and *Psa* C48 (NZ_CP032631.1) using Bowtie2 (version 2.3.4.3) (*Zhang et al., 2008*). Subsequently, binding peaks (q<0.01) were identified using MACS2 software (version 2.1.0). The enriched loci for each TF were annotated using the R package ChIPpeakAnno (version 3.18.2) (*Zhu et al., 2010*). We defined intergenic region before each TF sequence as the promoter region. As pHM1 plasmid takes its own promoter, we amplified the TF-CDS and cloned into the plasmid. The TF protein expression was activated by the promoter of plasmid.

## Electrophoretic mobility shift assay

DNA probes were amplified from *Psph* 1448A genome by PCR using primers listed in *Supplementary file 6*. The probes (20 ng) were mixed with various concentrations of proteins in 20 µl of gel shift buffer (10 mM Tris-HCl, pH 7.4, 50 mM KCl, 5 mM $MgCl_2$, 10% glycerol). After incubation at room temperature for 20 min, the samples were analysed by 6% native polyacrylamide gel electrophoresis (90 V for 90 min for sample separation). The gels were subjected to Gel Red dye (Tiangen Biotech) for 5 min and photographed by using a gel imaging system (Bio-Rad). The assay was repeated at least three times with similar results.

## RT-qPCR

The RT-qPCR primers used are shown in *Supplementary file 6* in the Supplemental Information. The cultured bacteria were grown to $OD_{600}$=0.6 and the total RNA were purified with Bacteria Total RNA Isolation Kit (Sangon Biotech). The RNA concentrations were measured using a NanoDrop 2000 spectrophotometer (Thermo Fisher). cDNA synthesis was performed using HiScript III RT SuperMix (Vazyme, China). RT-qPCR was performed with a SuperReal Premix Plus (SYBR Green) kit (Vazyme, China) according to the manufacturer's instructions. The reactions used 100 nM primers and were run for 40 cycles at 95°C for 30 s and 95°C for 10 s, and at 60°C for 30 s. The fold change represents relative expression level of mRNA, which can be estimated by the values of $2^{-(\Delta\Delta Ct)}$. The relative expression of target genes in WT was set to 1. All the reactions were conducted with three repeats.

## Motility assay

The motility assay was performed based on our previous study (*Shao et al., 2021*). Swimming plates were KB agar plates containing 0.3% agar (MP Biomedicals, UK) and rifampicin at 25 µg/ml. Overnight bacterial cultures were inoculated on swimming plates as 2 µl aliquots and incubated at room temperature for 3–5 days. Finally, the diameter of motility trace represented the swimming motility

of *P. syringae* strains. Photographs were taken by using the Bio-Rad imaging system. The assay was repeated at least three times with similar results.

### Congo red assay

Congo red assay was performed as previous study (*Shao et al., 2021*). Congo red plates were KB agar plates containing 1.0% agar (MP Biomedicals, UK) and rifampicin at 25 μg/ml. Overnight bacterial cultures were inoculated on Congo red plates as 2 μl aliquots and incubated at 28°C. The colony staining was photographed after 5–7 days. The assay was repeated at least three times with similar results.

### Biofilm formation assay

Biofilm formation assay was performed as previously described (*Shao et al., 2019*). Overnight bacterial cultures were transferred to a sterile 10 ml borosilicate tube containing 2 ml KB medium with rifampicin (25 μg/ml) with the original concentration $OD_{600}$=0.1. The cultures grew at room temperature for 36 hr without shaking. 0.1% crystal violet was used to stain the biofilm adhered to the tube for 20 min without shaking. Tubes were washed for more than three times with distilled deionised water gently, and other components on the tube were loosely washed off. The tubes were dry and photographed. The remaining crystal violet was fully dissolved in 1 ml 95% ethanol with constant shaking and its optical density at 590 nm was measured (BioTek microplate reader). The assay was repeated at least three times with similar results.

### Plant infection assay

Bean leaves were used for pathogenicity. Plant materials were grown in a greenhouse as described previously (*Pan et al., 2020*; *Xiao et al., 2007*). Overnight bacterial cultures were diluted to $OD_{600}$=0.2. The diluted cultures were washed three times by sterile water resuspended with equal volume sterile water and then continued to dilute 100-fold. The diluted bacterial solution was individually injected into bean leaves form stomata on the underside of a leaves. The plant was continued to grow at 28°C. Disease symptoms on bean leaves were measured 5 days after inoculation. Infected leaves (1 cm$^2$) were removed after 5 days growth and homogenised in sterile water. Bacteria were diluted to proper concentration and plated on a KB plate containing rifampicin at 25 μg/ml for bacterial count. The bacterial number represented colony-forming unit formed in one leaf disks (1 cm$^2$) with four repeats.

### Network and functional enrichment analysis

Network analyses were performed on Gephi 0.10. Functional annotations were retrieved from the *Pseudomonas* database and GO functional enrichment analyses, and KEGG analysis was performed using DAVID v6.8.

### Statistical analysis

Two-tailed Student's t-tests were performed using Microsoft Office Excel 2010. *p<0.05, **p<0.01, and ***p<0.001 and results represent means± SD. All experiments were repeated at least twice.

### Data availability

Sequencing data have been deposited and publicly available in Gene Expression Omnibus (GEO) under accession number GSE247395. Source data contain the numerical data used to generate the figures of EMSA. Analysis codes have been deposited as accession, https://github.com/dengxinb2315/PS-PATRnet-code (copy archived at *Deng, 2024*). Hierarchical information and functional categories of TFs are available in *Supplementary file 1*, *Supplementary file 2*, and *Supplementary file 3*. Evolutionary details are provided in *Supplementary file 4*. Primers and strains used in this paper are provided in *Supplementary file 6*.

## Acknowledgements

XD, YS, JW, and JH conceived the project. YS, JW, JH, SL, and YL carried out experiments. JH, SL, YL, and BL performed data analysis. XD, YS, JW, and JH wrote the paper. This study was supported by grants from the National Natural Science Foundation of China (32172358 to XD), General Research

Funds of Hong Kong (11101722, 11102223, and 11102720 to XD), Shenzhen Science and Technology Fund (JCYJ20210324134000002 to XD), Theme-based Research Scheme (T11-104/22R to XD), Guangdong Major Project of Basic and Applied Basic Research (2020B0301030005 to XD). The funders had no role in study design, data collection, interpretation, or the decision to submit the work for publication.

## Additional information

### Funding

| Funder | Grant reference number | Author |
| --- | --- | --- |
| National Natural Science Foundation of China | 32172358 | Xin Deng |
| General Research Funds of Hong Kong | 11101722 | Xin Deng |
| General Research Funds of Hong Kong | 11102223 | Xin Deng |
| General Research Funds of Hong Kong | 11102720 | Xin Deng |
| Shenzhen Science and Technology Fund | JCYJ20210324134000002 | Xin Deng |
| Theme-based Research Scheme | T11-104/22-R | Xin Deng |
| Guangdong Major Project of Basic and Applied Basic Research | 2020B0301030005 | Xin Deng |

The funders had no role in study design, data collection and interpretation, or the decision to submit the work for publication.

### Author contributions

Yue Sun, Methodology, Writing – original draft; Jingwei Li, Validation; Jiadai Huang, Software, Visualization; Shumin Li, Software; Youyue Li, Beifang Lu, Formal analysis; Xin Deng, Conceptualization, Resources, Supervision, Writing – review and editing

### Author ORCIDs

Yue Sun (ID) http://orcid.org/0009-0008-2843-0017
Jiadai Huang (ID) http://orcid.org/0000-0002-4104-6836
Youyue Li (ID) https://orcid.org/0000-0003-4976-3407
Xin Deng (ID) https://orcid.org/0000-0003-1580-0089

Reviewer #2 (Public Review): https://doi.org/10.7554/eLife.96172.3.sa1
Reviewer #3 (Public Review): https://doi.org/10.7554/eLife.96172.3.sa2
Author response https://doi.org/10.7554/eLife.96172.3.sa3

## Additional files

### Supplementary files

Supplementary file 1. Hierarchical height and direct interaction of transcription factors (TFs).

Supplementary file 2. Functional category and peak numbers of top transcription factors (TFs), functional category of middle and bottom TFs.

Supplementary file 3. Hierarchical network of transcription factors (TFs).

Supplementary file 4. Homologous proteins in three strains (DC3000, B728a, and C48) with 1448A and binding peaks of five TFs (Irp, X2193, X2193, X4127, and OmpR).

Supplementary file 5. List of orthologous genes of 1448A transcription factors (TFs) in the other tested *P. syringae* strains (DC3000, B728a, and C48).

Supplementary file 6. Strains and primers used in this study.

MDAR checklist

## Data availability

Sequencing data have been deposited and publicly available in Gene Expression Omnibus (GEO) under accession number GSE247395. Source data contain the numerical data used to generate the figures of EMSA. Analysis codes have been deposited as accession https://github.com/dengxinb2315/PS-PATRnet-code (copy archived at *Deng, 2024*).

The following dataset was generated:

| Author(s) | Year | Dataset title | Dataset URL | Database and Identifier |
| --- | --- | --- | --- | --- |
| Deng X | 2024 | Architecture of the genome-wide transcriptional regulatory network reveals the dynamic biological functions and divergent evolutionary trajectory in Pseudomonas syringae | https://www.ncbi.nlm.nih.gov/geo/query/acc.cgi?acc=GSE247395 | NCBI Gene Expression Omnibus, GSE247395 |

The following previously published dataset was used:

| Author(s) | Year | Dataset title | Dataset URL | Database and Identifier |
| --- | --- | --- | --- | --- |
| Fan L, Wang T, Hua C, Sun W, Li X, Grunwald L, Liu J, Wu N, Shao X, Yin Y, Yan J, Deng X | 2020 | The Compendium of DNA-Binding Specificities of Transcription Factors in a Pathogenic Bacterium | https://www.ncbi.nlm.nih.gov/geo/query/acc.cgi?acc=GSE146697 | NCBI Gene Expression Omnibus, GSE146697 |

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
