## [Editor Report · eLife assessment]

This work advances our understanding of transcriptional regulation of virulence and metabolic pathways in plant pathogenic bacteria. **Solid** evidence for the claims is provided by computational analysis of newly generated data on the genome-wide binding of 170 transcription factors to their target genes, together with experimental validation of the biological functions of some of these transcription factors. The findings and resources from this study will be **valuable** to researchers in the fields of systems biology, bacteriology, and plant-microbe interactions.

---

## [Referee Report · Reviewer #2 (Public Review)]

Summary:

The phytopathogenic bacterium Pseudomonas syringae is comprised of many pathovars with different host plant species and has been used as a model organism to study bacterial pathogenesis in plants. Transcriptional regulation is key to plant infection and adaptation to host environments by this bacterium. However, researches have focused on limited number of transcription factors (TFs) that regulate virulence-related pathways. Thus, a comprehensive, systems-level understanding of regulatory interactions between transcription factors in P. syringae has not been achieved.

This study by Sun et al performed ChIP-seq analysis of 170 out of 301 TFs in P. syringae pv. syringae 1448A and used this unique dataset to infer transcriptional regulatory networks in this bacterium. The network analyses revealed hierarchical interactions between TFs, various network motifs, and co-regulation of target genes by TF pairs, which collectively mediate information flow. As discussed, the structure and properties of the P. syringae transcriptional regulatory networks are somewhat different from those identified in humans, yeast, and *E. coli*, highlighting the significance of this study. Further, the authors made use of the P. syringae transcriptional regulatory networks to find TFs of unknown functions to be involved in virulence-related pathways. For some of these TFs, their target specificity and biological functions, such as motility and biofilm formation, were experimentally validated. Of particular interest is the finding that despite conservation of TFs between P. syringae pv. syringae 1448A, P. syringae pv. tomato DC3000, P. syringae pv. syringae B728a, and P. syringae pv. actinidiae C48, some of the conserved TFs show different repertoires of target genes in these four P. syringae strains.

Strengths:

This study presents a systems-level analysis of transcriptional regulatory networks in relation to P. syringae virulence and metabolism, highlights differences in transcriptional regulatory landscapes of conserved TFs between different P. syringae strains, and develops a user-friendly database for mining the ChIP-seq data generated in this study. These findings and resources will be valuable to researchers in the fields of systems biology, bacteriology, and plant-microbe interactions.

Weaknesses:

No major weaknesses were found, but some of the results may need to be interpreted with caution. ChIP-seq was performed with bacterial strains overexpressing TFs. This may cause artificial binding of TFs to promoters which may not occur when TFs are expressed at physiological levels. Another caution is applied to the interpretation of the biological functions of TFs during plant infection, as biological roles of the tested TFs are mostly based on in vitro experiments.

This work advances our understanding of transcriptional regulation of virulence and metabolic pathways in plant pathogenic bacteria. Solid evidence for the claims is provided by computational analysis of newly generated data on the genome-wide binding of 170 transcription factors to their target genes, together with experimental validation of the biological functions of some of these transcription factors. The findings and resources from this study will be valuable to researchers in the fields of systems biology, bacteriology, and plant-microbe interactions.

---

## [Referee Report · Reviewer #3 (Public Review)]

Summary:

This study aims to understand gene regulation of the plant bacterial pathogen Pseudomonas syringae. Although the function of some TFs has been characterized in this strain, a global picture of the gene regulatory network remains elusive. The authors conducted a large-scale ChIP-seq analysis, covering 170 out of 301 TFs of this strain, and revealed gene regulatory hierarchy with functional validation of some previously uncharacterized TFs.

Strength:

- This study provides one of the largest ChIP-seq datasets for a single bacterial strain, covering more than half of its TFs. This impressive resource enabled comprehensive systems-level analysis of the TF hierarchy.

- This study identified novel gene regulation and function with validations through biochemical and genetic experiments.

- The authors conducted broad analyses including comparisons between different bacterial strains, providing further insights into the diversity and conservation of gene regulatory mechanisms.

---

## [Author Response]

The following is the authors’ response to the original reviews.

**Public Reviews:**

**Reviewer #1 (Public Review):**
In this work, the authors provide a comprehensive description of transcriptional regulation in Pseudomonas syringae by investigating the binding characteristics of various transcription factors. They uncover the hierarchical network structure of the transcriptome by identifying top-, middle-, and bottom-level transcription factors that govern the flow of information in the network. Additionally, they assess the functional variability and conservation of transcription factors across different strains of P. syringae by studying DNA-binding characteristics. These findings notably expand our current knowledge of the P. syringae transcriptome.The findings associated with crosstalk between transcription factors and pathways, and the diversity of transcription factor functions across strains provide valuable insights into the transcriptional regulatory network of P. syringae. However, these results are at times underwhelming as their significance is unclear. This study would benefit from a discussion of the implications of transcription factor crosstalk on the functioning of the organism as a whole. Additionally, the implications of variability in transcription factor functions on the phenotype of the strains studied would further this analysis.Overall, this manuscript serves as a key resource for researchers studying the transcriptional regulatory network of P. syringae.

Thank you for your positive comments.

**Reviewer #2 (Public Review):**

Summary:The phytopathogenic bacterium Pseudomonas syringae is comprised of many pathovars with different host plant species and has been used as a model organism to study bacterial pathogenesis in plants. Transcriptional regulation is key to plant infection and adaptation to host environments by this bacterium. However, researchers have focused on a limited number of transcription factors (TFs) that regulate virulence-related pathways. Thus, a comprehensive, systems-level understanding of regulatory interactions between transcription factors in P. syringae has not been achieved.This study by Sun et al performed ChIP-seq analysis of 170 out of 301 TFs in P. syringae pv. syringae 1448A and used this unique dataset to infer transcriptional regulatory networks in this bacterium. The network analyses revealed hierarchical interactions between TFs, various network motifs, and co-regulation of target genes by TF pairs, which collectively mediate information flow. As discussed, the structure and properties of the P. syringae transcriptional regulatory networks are somewhat different from those identified in humans, yeast, and *E. coli*, highlighting the significance of this study. Further, the authors made use of the P. syringae transcriptional regulatory networks to find TFs of unknown functions to be involved in virulence-related pathways. For some of these TFs, their target specificity and biological functions, such as motility and biofilm formation, were experimentally validated. Of particular interest is the finding that despite conservation of TFs between P. syringae pv. syringae 1448A, P. syringae pv. tomato DC3000, P. syringae pv. syringae B728a, and P. syringae pv. actinidiae C48, some of the conserved TFs show different repertoires of target genes in these four P. syringae strains.

Thank you for your positive comments.

Strengths:This study presents a systems-level analysis of transcriptional regulatory networks in relation to P. syringae virulence and metabolism, and highlights differences in transcriptional regulatory landscapes of conserved TFs between different P. syringae strains, and develops a user-friendly database for mining the ChIP-seq data generated in this study. These findings and resources will be valuable to researchers in the fields of systems biology, bacteriology, and plant-microbe interactions.

Thank you for your positive comments.

Weaknesses:No major weaknesses were found, but some of the results may need to be interpreted with caution. ChIP-seq was performed with bacterial strains overexpressing TFs. This may cause artificial binding of TFs to promoters which may not occur when TFs are expressed at physiological levels. Another caution is applied to the interpretation of the biological functions of TFs. The biological roles of the tested TFs are based on in vitro experiments. Thus, functional relevance of the tested TFs during plant infection and/or survival under natural environmental conditions remains to be demonstrated.

Thank you for your comments, and we agree with the reviewer. To eliminate the artificial binding of TFs, we performed EMSA to verify the analyzed targets. Our EMSA results confirmed the analyzed binding peaks.

For the verification experiments of the biological functions of TFs, we also performed in vivo motility assay and biofilm production assay (Figures 3b-d). To further detect the biological functions of TFs, we performed plant infection assay of TF PSPPH2193 under natural environmental condition (bean leaves). As shown in Figures S6c and g, both the motility and the virulence of P. syringae in ∆PSPPH2193 strain was significantly reduced compared with WT strain. These results showed that TF PSPPH2193 positively regulated the pathogenicity of P. syringae via modulating the bacterial motility.

**Reviewer #3 (Public Review):**
Summary:This study aims to understand gene regulation of the plant bacterial pathogen Pseudomonas syringae. Although the function of some TFs has been characterized in this strain, a global picture of the gene regulatory network remains elusive. The authors conducted a large-scale ChIP-seq analysis, covering 170 out of 301 TFs of this strain, and revealed gene regulatory hierarchy with functional validation of some previously uncharacterized TFs.

Thank you for your positive comments.

Strengths:- This study provides one of the largest ChIP-seq datasets for a single bacterial strain, covering more than half of its TFs. This impressive resource enabled comprehensive systems-level analysis of the TF hierarchy.- This study identified novel gene regulation and function with validations through biochemical and genetic experiments.- The authors attempted on broad analyses including comparisons between different bacterial strains, providing further insights into the diversity and conservation of gene regulatory mechanisms.

Thank you for your positive comments.

Weaknesses:(1) Some conclusions are not backed by quantitative or statistical analyses, and they are sometimes overinterpreted.

Thank you for your comments. We used hypergeometric test in this analysis. Although only one gene was enriched in some pathways, the adjusted p-value was less than 0.05. We added the details in the revised manuscript.

(2) Some figures and analyses are not well explained, and I was not able to understand them.

Thank you for your comments, and we are sorry for the confusion. We defined ‘indirect interaction’ as ‘co-association’ and ‘cooperativity’ as ‘if the common target of two TFs is from a TF’. We added the definition of "indirect interaction" and "cooperativity" in the revised legend.

For Figure S3a, the low co-association scores and large peak numbers of these top-level TFs indicated that top-level TFs preferred to solely regulate target genes, but not to co-regulate with other top-level TFs. PSPPH4700 was an example to show that top-level TFs with low co-association scores and large peak numbers tend to solely regulate target genes, but not to co-regulate with other top-level TFs. We revised the sentence to ‘For example, the top-level TF PSPPH4700 yielded over 1,700 peaks but cooperated with only 24 top-level TFs with low co-association scores about 0.05 (Supplementary Table 2b).’.

We analyzed high co-association scores of 125 TFs in three levels and further determined the co-association patterns. To identify the tendency of co-association of all these 125 TFs, the co-association patterns were classified into 4 clusters. Bottom-level TFs tend to co-regulate target genes with other TFs. We revised the sentence in the revised manuscript.

For Figure 2b, in C1, C2 and C4, many bottom-level TFs performed co-association pattern with other TFs, especially bottom TFs (showed in C4). To explore the regulatory pattern in C3, the peak locations in target genes of MexT were analyzed with those of TFs in C3. Seven top-level TFs (PSPPH1435, PSPPH1758, PSPPH2193, PSPPH2454, PSPPH4638, PSPPH4998 and PSPPH3411), three middle-level TFs (PSPPH1100, PSPPH5132 and PSPPH5144) and four bottom-level TFs (PSPPH0700, PSPPH2300, PSPPH2444 and PSPPH2580) were compared with MexT. MexT showed higher co-association scores (more than 60 scores) with more top-level-TFs. Therefore, we demonstrated that MexT performed closer co-association relationships with top-level TFs. We added the statement in the revised manuscript.

For Figure 1a, the hierarchical network showed different number of TFs in three levels (54 top-level TFs, 62 middle-level TFs and 147 bottom-level TFs), which indicated that more than half of TFs (bottom-level TFs) tend to be regulated by other TFs and then directly bound to target genes. This finding showed a downward regulatory direction of transcription regulation in P. syringae. We revised the statement in the revised manuscript.

(3) The Method section lacks depth, especially in data analyses. It is strongly recommended that the authors share their analysis codes so that others can reproduce the analyses.

Thank you for your comments, and we defined the intergenic region before each TF sequence as the promoter region. As pHM1 plasmid carries its own constitutive promoter (lacZ promoter), we amplified the TF-coding sequence and cloned into site following the promoter. The TF protein expression was activated by the promoter of plasmid. Psph 1448A was used for our main ChIP-seq. We added the details in the revised manuscript.

For Figure S3, we performed GO analysis on genes that were co-bound by TF pairs. We added the details in the revised manuscript.

We shared our analysis codes on the website (https://github.com/dengxinb2315/PS-PATRnet-code) in the Data Availability.

Recommendations for the authorsReviewer #1 (Recommendations For The Authors):(1) The specific strain of Pseudomonas syringae used in the study outside of the evolutionary analysis should be specified in the abstract and main text.

Thank you for your suggestion. We revised the statements in abstract and main text to specific strains.

(2) The language used throughout the manuscript should be revised for clarity, conciseness, and readability.

Thank you for your suggestion. We have revised the language used throughput the manuscript by a scientific editor who is a native speaker of English.

(2) Line 688: Replace "80C" with "-80C".

Thank you for your correction. We revised ‘80℃’ to ‘-80℃’. Please see Line 713.

(3) Line 172 - 173: The abbreviations TT, MM, BB, TM, TB, and MB need to be expanded in the main text before their use.

Thank you for your suggestion. We added the abbreviations TT, MM, BB, TM, TB, and MB in the manuscript. Please see Lines 172-174.

**Reviewer #2 (Recommendations For The Authors):**
Major points(1) The name of the P. syringae strains used in each experiment/analysis should be explicitly stated (most experiments were carried out with P. syringae strain 1448A). This should also be applied to the introduction where many papers on P. syringae are cited without clear indication of strain names. I think this amendment is essential because target genes and thus biological functions of TFs could be different between P. syringae strains, as shown in the present study.

Thank you for your suggestion. We revised the P. syringae strains in the citations throughout the manuscript.

(2) How many TFs were analyzed throughout the study? Most sentences including line 22 in the abstract say 170, but I also found some say 270 (for example, line 106 and line 149). The legend of Figure 1 says 262. More detailed information is required regarding the datasets used for each analysis.

Thank you for your suggestion. The number of TFs analyzed by ChIP-seq in this research is 170, the number of TFs analyzed by HT-SELEX in our previous research is 100. Hierarchical analysis integrated data from ChIP-seq and HT-SELEX which included 270 TFs. As 8 TFs did not show hierarchical characteristic, the legend of Figure 1 said 262 TFs. We added the data source in the revised manuscript. Please see Lines 104, 147, 160 and 1082.

(3) Figure 1b: Please define "indirect interaction" and "cooperativity" in the legend as well as in the text. I only found the definition of "direct interaction".

Sorry for the missing information. We defined ‘indirect interaction’ and ‘cooperativity’ as ‘co-association’ and ‘if the common target of two TFs is from a TF’, respectively. We added the definition of "indirect interaction" and "cooperativity" in the revised legend. Please see Lines 174-176, 1084-1086.

(4) I found it very interesting that conserved TFs show different repertoires of target genes in different P. syringae strains. This suggests the rewiring of transcriptional regulatory networks in P. syringae strains, but the underlying mechanism is not explored in the current manuscript. It can be easily tested whether these conserved TFs bind to similar or different motifs by motif enrichment analysis. If they bind to similar motifs, it is possible that the promoter sequences of their target genes have diversified. Addressing or at least discussing these points would provide molecular insights into the diversification of the transcriptional regulatory networks in P. syringae. Similarly, functional enrichment analysis of target genes can be used to test whether the conserved TFs regulate different biological processes.

Thank you for your suggestion. We added the motif analysis and functional enrichment analysis of target genes of TFs (PSPPH3122 and PSPPH4127) in different P. syringae strains. We found two different motifs (AGACN4GATCAA and CGGACGN3GATCA) in 1448A and DC3000 strains, respectively. We also performed the GO analysis and found the specific functions of PSPPH3122 in Psph 1448A compared with Pst DC3000 and Pss B728a strains, including recombinase activity and DNA recombination. For PSPPH4127, we found four different motifs in four P. syringae strains. GO analysis showed its relationship with recombinase activity in Psph 1448A strain, and RNA binding, structural constituent of ribosome, translation and ribosome in Pss B728a strain. These results indicated the highly functional diversity of TFs in P. syringae. We added these points in the Results part, and Figure S9-S10 in the revised manuscript. Please see Lines 497-509.

(5) Related to point 4, it would be quite useful if a list of orthologous genes of 1448A TFs in the other tested P. syringae strains were provided. Such information may also enhance the utility of the database developed in this study.

Thank you for your suggestion. We added the list of orthologous genes of 301 Psph 1448A TFs in the other tested P. syringae strains in the Supplementary Table 5. Please see Lines 467 and Supplementary Table 5.

(6) Lines 243-246: It is unclear how these functional enrichment analyses were performed. Did you use target genes regulated by individual TFs or those coregulated by pairs of TFs? Please add more information for the sake of readers.

Thank you for your suggestion. We performed the functional enrichment analyses by hypergeometric test (BH-adjusted p < 0.05) via using target genes regulated by individual TFs. We added the details in the Results part. Please see Lines 248-252, 270, 1194-1195, 1199-1200 and 1205-1206.

Minor points(1) Lines 167-168: I may not understand correctly, but you might want to say "downward-pointing edges" instead of "upward-pointing edges".

Thank you for correction. We revised the ‘upward-pointing edges’ to ‘downward-pointing edges’. Please see Line 166.

(2) Line 174: "physical interactions" should be amended to "direct interactions".

Thank you for correction. We revised the ‘physical interactions’ to ‘direct interactions’. Please see Line 177.

(3) Line 224: Could you please explain why bacterial growth in plant tissues is considered an example of "multi-stability"?

Thank you for your suggestion. We are sorry for the incorrect statement. We showed ‘plant intercellular spaces’ as ‘multi-stability’. We revised the sentence to ‘These auto-regulators are important and always act as repressors in scenarios of multi-stability, such as plant intercellular spaces’. Please see Lines 224-226.

(4) Line 254-257: Here, the definition of "tether binding" is introduced, but it is not very clear to me. In my understanding, tethered binding is an indirect binding of a TF to a target gene through protein-protein interaction with other TF that directly binds to the promoter of the target gene.

Thank you for your suggestion, and we agree with you. We referred to the paper published in 2012 (Wang et al., 2012) and revised the statement of ‘tether binding’ to ‘This finding suggested that these TFs indirectly regulated target genes through protein-protein interaction with other TFs that directly binds to the promoters of target genes, a phenomenon defined as tethered binding’. Please see Lines 259-262.

(5) Lines 341-343: Figure 3b shows qRT-PCR of hopAE1, not hrpR.

Thank you for your correction. We revised ‘hrpR’ to ‘hopAE1’. Please see Line 349.

(6) Lines 500 and Figure 6b: It is hard to see edges from module 12 to others. So, it would be better to provide numeric information (number of TFs and target genes) in the text.

Thank you for your suggestion. Module 12 includes 22 TFs and 318 target genes. We added the statement of numeric information about Module 12 in the revised manuscript. Please see Lines 536-537.

(7) Line 519: Figure S4b is not the EMSA data for PSPPH3798. Should it be Figure S4e?

Thank you for your correction. We revised to ‘Figure S4e’. Please see Line 545.

(8) Line 522: Figure S6b is not relevant to the statement here.

Thank you for your correction. We deleted the ‘Figure S6b’ here. Please see Line 547.

(9) Line 593: prokaryotic transcriptional regulatory networks -> eukaryotic transcriptional regulatory networks?

Thank you for your correction. We revised ‘prokaryotic transcriptional regulatory networks’ to ‘eukaryotic transcriptional regulatory networks’. Please see Line 618.

(10) Figure S3 requires images of higher resolution. Especially, values for the color codes are not readable or very hard to see.

Thank you for your suggestion. To make the images clearer, we enlarged the images, change the color codes, and divided it into three figures. Please see the revised Figures S3-S5 and corresponding Figure legends at Lines 1191-1206.

**Reviewer #3 (Recommendations For The Authors):**(1) Some conclusions are not backed by quantitative or statistical analyses, and they are sometimes overinterpreted.L221: "Taken together, the simplest and most effective submodule M1 and the coregulatory submodule M13 played crucial roles in the transcriptional regulation of TFs in P. syringae."The authors did not provide any evidence supporting the functional importance of any of these submodules. M13 is most enriched within the locked loop, but its size is much smaller than simple loops. What evidence supports the importance of this particular submodule?

Thank you for your suggestion. In eukaryote (*Saccharomyces cerevisiae*) and prokaryote (*Escherichia coli*) which have the best characterized transcriptional regulation networks, the feed-forward loop (called M13 in this article) appear numerous times in the networks and perform different biological functions. M1 appeared most frequently by an order of magnitude than other modules. We revised the sentence to ‘Taken together, the most numerous but simplest submodule M1 played a crucial role in the transcriptional regulation of TFs in P. syringae.’ Please see Lines 222-224.

L223: "...we found 92 auto-regulators...These auto-regulators are important and always act as repressors in scenarios of multi-stability, such as in plant intercellular spaces where bacteria grow (Figure 1d)(Alon, 2007). These regulators are regarded as bistable switches that further influence the expression of downstream genes."Are these claims supported by any evidence?

Thank you for your suggestion. We referred to the following articles:

(1) Alon. Nature Reviews Genetics. 2007(Alon, 2007).

That transcription factors repress the transcription of their target genes was considered as negative regulation. These negative autoregulators account for half of the repressors in *E. coli* and occur in many eukaryotes. The repressors controlled the concentration of the target production through suppressing its expression, which accelerated back to the steady state of cells.

(2) Becskei. et al. Nature. 2000; Rosenfeld et al. Journal of Molecular Biology. 2002 (Becskei & Serrano, 2000; Rosenfeld, Elowitz, & Alon, 2002).

Fluorescent assay confirmed that the negative autoregulatory module (negative autoregulator TetR) spent less time to the log phase than unregulated group, which reduced cell-to-cell fluctuations in the steady-state level of the transcription factor. Some negative autoregulators were showed here, such as LexA, CysB and SrlA-D.

In our research, we also identified many autoregulators including CysB and LexA2 (annotated as LexA repressor). We revised the sentence to ‘In addition, we found 92 auto-regulators in our hierarchy network. These auto-regulators are important and always act as repressors in scenarios of multi-stability, such as plant intercellular spaces (Figure 1d) (Alon, 2007). For example, LexA and CysB as negative autoregulators were indicated to reduce cell-to-cell fluctuations in the steady-state level of the transcription factor (Becskei & Serrano, 2000; Rosenfeld et al. 2002).’. Please see Lines 224-229.

L265: "This finding indicated that the bottom-level TFs, which were more easily regulated, tended to cooperate with downstream genes and other intra-level TFs."Could the authors provide more explanation to reach this conclusion from the data? Analyzing the number of highly co-accessing TFs does not sufficiently support this conclusion. The clustering of TFs (C1-C4) is incomplete, and each TF level (Top/Middle/Bottom) contains different numbers of TFs. Since the authors calculated all-by-all co-association scores for these 125 TFs, they can group these scores into 6 possible combinations (TT, TM, TB, MM, MB, BB) and show the distribution of co-association scores.

Thank you for your suggestion. We indicated that the bottom-level TFs preferred to regulate the target genes through the cooperation with other TFs. To further support the claim, we analyzed the proportion of the bottom TF interaction in all the TF pairs interactions and direct interaction based on results in Figure 1B. The interactions of bottom TFs were 43% and 49%, respectively. However, the interactions of top TFs and middle TFs were only 20% and 28%, respectively. We revised the statement ‘Based on the analysis in Figure 1B, we found that the proportions of bottom-level TF interaction in all the TF pair interactions and direct interaction were 43% and 49%. These results indicated that the bottom-level TFs tended to regulate downstream genes through cooperating with other level TFs.’ in the revised manuscript. Please see Lines 269-272.

As not every TF performed co-association with other TFs, we only collected 125 TFs with co-association scores. For the numbers of TF in each level, we divided TFs into three levels according to hierarchy height. Hierarchy height from -1 to -0.3 represented bottom level; hierarchy height from -0.3 to 0.3 represented middle level ; hierarchy height from 0.3 to 1 represents top level. Each level was equally divided by height scores. We suggested that different numbers of TFs in three levels indicated the characteristic of transcriptional regulation in P. syringae.

Thank you for your suggestion. As the co-association patterns were determined by co-association scores of the same TFs, we first grouped the co-association scores into 3 possible TF pairs (TT, MM, and BB, in Figures S3a, S4a and S5a). Our results indicated that higher co-association scores preferred to occur in bottom-level TFs. We revised the statement in the revised manuscript. Please see Lines 244-252.

(2) Some figures and analyses are not well explained, and I was not able to understand them.Figure 1b: The terms "direct," "indirect," and "cooperativity" require further clarification as their definitions in the text (L169-183) are unclear to me. This ambiguity hampers the evaluation of the authors' discussion regarding TF-TF interactions (L561-584), an important theme of this study. The figure includes concepts discussed in later sections (e.g., cooperativity), making it difficult to understand. A diagram explaining these concepts would be highly helpful for readers to understand.

Sorry for the missing information. We defined ‘indirect interaction’ as ‘co-association’, ‘cooperativity’ as ‘if the common target of two TFs is from a TF’. We added the definition of "indirect interaction" and "cooperativity" in the revised manuscript and legend. Please see Lines 174-176 and 1085-1087.

L253: "Notably, we found that TFs at the top level, without cooperating TFs, exhibited a large number of binding peaks (Figure S3a)."

I could not understand this sentence. Did the authors mean that top-level TFs with a large number of peaks showed a low level of co-association? If so, does this data suggest that these TFs do not tend to cooperate with other TFs? I was confused by the discussion in L253-L261.

Thank you for your comment, and we agree with you. The low co-association scores and large peak numbers of these top-level TFs indicated that top-level TFs preferred to solely regulate target genes, but not to co-regulate with other top-level TFs.

Thank you for your comment. From L253-256, PSPPH4700 was an example to show that top-level TFs with low co-association scores and large peak numbers tend to solely regulate target genes, but not to co-regulate with other top-level TFs. We revised the sentence to ‘For example, the top-level TF PSPPH4700 yielded over 1,700 peaks, but cooperated with only 24 top-level TFs with low co-association scores about 0.05 (Supplementary Table 2b).’.

From L257-261, we analyzed high co-association scores of 125 TFs in three levels and further determined the co-association patterns. To identify the tendency of co-association of all these 125 TFs, the co-association patterns were classified into 4 clusters. Bottom-level TFs tend to co-regulate target genes with other TFs. We revised the sentence. Please see Lines 262-264, 265-266 and 269-272.

L287: "The analysis of the peak locations of MexT demonstrated that MexT showed closer co-association relationships with top-level TFs (Figure 2b)."I could reach this conclusion by seeing Figure 2b. Additional explanation and/or data visualization would be appreciated.

Thank you for your suggestion. In C1, C2 and C4, many bottom-level TFs performed co-association pattern with other TFs, especially bottom TFs (showed in C4). To explore the regulatory pattern in C3, the peak locations in target genes of MexT were analyzed with those of TFs in C3. Seven top-level TFs (PSPPH1435, PSPPH1758, PSPPH2193, PSPPH2454, PSPPH4638, PSPPH4998 and PSPPH3411), three middle-level TFs (PSPPH1100, PSPPH5132 and PSPPH5144) and four bottom-level TFs (PSPPH0700, PSPPH2300, PSPPH2444 and PSPPH2580) were compared with MexT. MexT showed higher co-association scores (more than 60 scores) with more top-level-TFs. Therefore, we demonstrated that MexT performed closer co-association relationships with top-level TFs. We added the statement in the revised manuscript. Please see Lines 291-296.

Figure 6cd: What kind of enrichment analysis did the authors perform? Was any statistical test used? The figure only shows the number of genes, and sometimes the number is only 1 for a functional category. Can it be considered as significant enrichment?

Thank you for your comment. We used hypergeometric test in this analysis. Although only one gene was enriched in some pathways, the adjusted p-value was less than 0.05. We added the details in the revised manuscript. Please see Lines 533-534.

L169: "The hierarchical network revealed a downward information flow, suggesting the prioritization of collaboration between different hierarchy levels."Can the authors please explain the logic behind this statement more in detail?

Thank you for your comment. The hierarchical network showed different number of TFs in three levels (54 top-level TFs, 62 middle-level TFs and 147 bottom-level TFs), which indicated that more than half of TFs (bottom-level TFs) tend to be regulated by other TFs and then directly bound to target genes. This finding showed a downward regulatory direction of transcription regulation in P. syringae. We revised the statement in the revised manuscript. Please see Lines 167-170.

(3) The Method section lacks depth, especially on data analyses.How did the authors define promoter regions of each gene? How were operons treated in their analyses? Was P. syringae 1448A used for their main ChIP-seq?

Thank you for your comment. We defined the intergenic region before each TF sequence as the promoter region.

As pHM1 plasmid carries its own constitutive promoter (lacZ promoter), we amplified the TF-coding sequence and cloned into the site following the promoter. The TF protein expression was activated by the promoter of plasmid.

P. syringae 1448A was used for our main ChIP-seq. We added the details in the revised manuscript. Please see Lines 705 and 727-730.

Figure S3: I am not sure how the GO analyses were done. For example, in the case of the top-level TF PSPPH4700, did the authors perform GO analysis on genes that are co-bound by PSPPH4700 and any other top-level TFs?

Thank you for your comment and we agree with you. We performed GO analysis on genes that were co-bound by TF pairs in the same level. We added the details in the revised manuscript. Please see Lines 248-252.

The analysis presented in Figure 6a needs more explanation of the methodology employed by the authors.

Thank you for your comment. We added more details for the analysis in Figure 6a. Please see Lines 514-522.

It is strongly recommended that the authors share their analysis codes so that others can reproduce the analyses.

Thank you for your comment. We shared our analysis codes on the website (https://github.com/dengxinb2315/PS-PATRnet-code) in the Data Availability. Please see Lines 800-801.

(4) Other:Figure 3: I suggest putting additional panel labels to facilitate the interpretation of the figure.

Thank you for your suggestion. We added detailed labels in the revised Figures 3 and 4. Please see in the revised Figures 3 and 4.

I spotted several potential errors:L106: 170 TFs?

Thank you for your comment, and we are sorry for the missing details. For the hierarchical network, we integrated the DNA-binding data of 170 TFs in this study and 100 TFs in our previous SELEX research. We added the details in the revised manuscript. Please see Lines 104, 147 and 159-160.

L592: P. syringae not *E. coli*?

Thank you for your comment. Here we discussed the hierarchical characteristics in *E. coli*. We revised the statement in the revised manuscript. Please see Line 618.

L593: eukaryotic not prokaryotic?

Thank you for your correction. Here we discussed the feedforward loops in our study. We revised the statement in the revised manuscript. Please see Line 618.

References

Alon, U. (2007). Network motifs: theory and experimental approaches. Nature Reviews Genetics, 8(6), 450-461.

Becskei, A., & Serrano, L. (2000). Engineering stability in gene networks by autoregulation. Nature, 405(6786), 590-593.

Rosenfeld, N., Elowitz, M. B., & Alon, U. (2002). Negative autoregulation speeds the response times of transcription networks. Journal of molecular biology, 323(5), 785-793.

Wang, J., Zhuang, J., Iyer, S., Lin, X., Whitfield, T. W., Greven, M. C., . . . Cheng, Y. (2012). Sequence features and chromatin structure around the genomic regions bound by 119 human transcription factors. Genome research, 22(9), 1798-1812.